**PLOS** | **ONE**

# Elucidating stygofaunal trophic web interactions via isotopic ecology

**Mattia Saccò**[1]*, **Alison J. Blyth**[1], **William F. Humphreys**[2,3], **Alison Kuhl**[4], **Debashish Mazumder**[5], **Colin Smith**[6], **Kliti Grice**[1]

**1** WA-Organic Isotope Geochemistry Centre, The Institute for Geoscience Research, School of Earth and Planetary Sciences, Curtin University, Bentley, WA (Australia), **2** Collections and Research Centre, Western Australian Museum, Welshpool, 6986, WA (Australia), **3** School of Biological Sciences, University of Western Australia, Crawley, Western Australia (Australia), **4** Organic Geochemistry Unit, Bristol Biogeochemistry Research Centre, School of Chemistry, University of Bristol, Bristol, United Kingdom, **5** Australian Nuclear Science and Technology Organisation (ANSTO), Locked Bag 2001, Kirrawee DC, NSW (Australia), **6** Department of Archaeology and History, La Trobe University, Bundoora, VIC (Australia)

* mattia.sacco@postgrad.curtin.edu.au

**Data Availability Statement:** All relevant data are within the manuscript and its Supporting Information files.

**Funding:** This research was funded by an Australian Research Council (ARC) Linkage grant

## Abstract

Subterranean ecosystems host highly adapted aquatic invertebrate biota which play a key role in sustaining groundwater ecological functioning and hydrological dynamics. However, functional biodiversity studies in groundwater environments, the main source of unfrozen freshwater on Earth, are scarce, probably due to the cryptic nature of the systems. To address this, we investigate groundwater trophic ecology *via* stable isotope analysis, employing $\delta^{13}C$ and $\delta^{15}N$ in bulk tissues, and amino acids. Specimens were collected from a shallow calcrete aquifer in the arid Yilgarn region of Western Australia: a well-known hot-spot for stygofaunal biodiversity. Sampling campaigns were carried out during dry (low rainfall: LR) and the wet (high rainfall: HR) periods. $\delta^{13}C$ values indicate that most of the stygofauna shifted towards more $^{13}C$-depleted carbon sources under HR, suggesting a preference for fresher organic matter. Conversion of $\delta^{15}N$ values in glutamic acid and phenylalanine to a trophic index showed broadly stable trophic levels with organisms clustering as low-level secondary consumers. However, mixing models indicate that HR conditions trigger changes in dietary preferences, with increasing predation of amphipods by beetle larvae. Overall, stygofauna showed a tendency towards opportunistic and omnivorous habits—typical of an ecologically tolerant community—shaped by bottom-up controls linked with changes in carbon flows. This study provides baseline biochemical and ecological data for stygofaunal trophic interactions in calcretes. Further studies on the carbon inputs and taxa-specific physiology will help refine the interpretation of the energy flows shaping biodiversity in groundwaters. This will aid understanding of groundwater ecosystem functioning and allow modelling of the impact of future climate change factors such as aridification.

## Introduction

During recent decades, investigations of trophic webs have become a cornerstone for the interpretation of functional biodiversity in freshwater ecosystems. Within both lentic and lotic

(LP140100555) to the University of Adelaide, Curtin University, and Flinders University, with industry partners, the Western Australian Museum, the South Australian Museum, Rio Tinto, Biota Environmental Sciences, Bennelongia Environmental Consultants and the Department of Parks and Wildlife (WA). Saccò is supported by a Curtin International Postgraduate Research Scholarship (CIPRS) and an AINSE postgraduate scholarship (PGRA). Blyth acknowledges an AINSE Research Fellowship (2012-2018).

**Competing interests:** The authors have declared that no competing interests exist.

environments, macroinvertebrate food web dynamics play a key role in shaping process-level aquatic ecosystem attributes [1]. Aquatic faunal trophic characterization is usually conducted by employing the morpho-behavioural based concept of functional feeding groups (FFGs) [2]. Since its inception, FFGs have been extensively used in ecological assessments and biomonitoring studies, and have allowed detailed assessment of ecological patterns in both natural and disturbed environments [3,4,5].

However, despite the hydraulic and ecological continuum in groundwater dependent ecosystems, the subsurface ecosystem and the study of its food chain interactions have suffered from a conceptual disconnection from surficial aquatic habitats. The main reasons are attributable to methodological limitations [6,7], scarce aquifer accessibility [8] and the lack of interdisciplinary approaches [9]. Moreover, compared to surface freshwater ecosystems, groundwaters are subjected to relatively extreme environmental conditions: sparse organic inputs, lack of light and primary production, and truncated trophic webs [10,11,12,13]. Altogether, these unique conditions shape obligate subterranean aquatic communities (stygofauna) dominated by plastic and opportunistic trophic behaviours [14,15], whose categorization *via* feeding modes such as FFGs is constantly at risk of misinterpretation. As a result, our knowledge about how food web interactions shape groundwater ecological functioning and community patterns is fragmented [16].

Stygofauna—when present—play a key role in regulating both ecological and hydrological dynamics in aquifers [17,18]: they actively bioturbate the sediment, facilitate nutrient recycling and, in combination with microbial communities, degrade/retain contaminants. In groundwaters, carbon inputs (allochthonous dissolved organic carbon (DOC) and chemoautotrophic production) are mediated by microbes which are then grazed by basal stygofauna [19]. Organic matter (OM) is transferred along the trophic chain *via* prey-predator interactions. Therefore, OM inputs, microbial communities, and the stygofaunal trophic web, all shape the energy flows sustaining the subterranean biodiversity [20].

The incorporation of biogeochemical approaches (i.e. stable isotopes composition, fatty acids content, radiocarbon analysis) has recently led to re-evaluation of the archetype of poorly structured–and generalist-dominated–trophic dynamics in groundwaters [21]. These designs are leading a vital transition from purely descriptive to functionally-based investigations, providing wider perspectives to the field [22].

Carbon ($\delta^{13}C$) and nitrogen ($\delta^{15}N$) stable isotope analysis (SIA) is a well-established approach enabling quantitative investigation of food webs [23,24]. Since its initial application in groundwater trophic ecology, several studies have benefited from the insights provided by the study of naturally-occurring stable isotopes [25,26]. However, $\delta^{13}C$ and $\delta^{15}N$ SIA investigations on bulk material are limited by the mixing of tissues and different biochemical pathways [27]. These limitations can be addressed by the complementary or alternative use of compound-specific approaches.

$\delta^{13}C$ and $\delta^{15}N$ Compound Specific Isotope Analysis (CSIA) on amino acids allows detailed characterization of food web interactions [28], by focusing on compounds created by definable biosynthetic pathways. Single amino acids can be divided into essential (EAA) and non-essential (NEAA). Whilst primary producers (plants, algae and bacteria) biosynthetise *de novo* EAA from a bulk carbon pool, animals lack these enzymatic pathways and acquire EAA from their diet [29]. As a result, tracking of EAA allows carbon fingerprinting of food sources down to the base of food webs [30]. Concurrently, $\delta^{15}N$ CSIA can distinguish between compounds reflecting the source isotopic signal, and that enriched with each trophic step, thus providing crucial information on prey-predator interactions [31]. The application of CSIA in amino acids has allowed a much more thorough understanding of food web dynamics in freshwater [32], marine [33] and terrestrial environments [34], but despite the greater potential than bulk

analysis [35], this technique has yet to be applied to food web studies of groundwater environments.

This study is, to our best knowledge, the first based on the combination of carbon and nitrogen CSIA in groundwaters, and focuses on a calcrete stygofaunal community under two contrasting environmental conditions: low rainfall (LR, dry season) and high rainfall (HR, wet season). We hypothesise that different environmental conditions trigger species-specific adaptations that are ultimately responsible for distinct food web interactions. The specific objectives of this work are: 1) unravel OM incorporation trends across the stygofaunal community, 2) decipher the trophic habits of the species and elucidate prey-predator interactions, and 3) provide biochemically-based knowledge about trophic web interactions in arid zone calcrete aquifers.

## Methodology

### Study area and field work

The field work was carried out at a calcrete aquifer (28˚41‘S 120˚ 58‘E) located on Sturt Meadows pastoral station, Western Australia, ~42 km from the settlement of Leonora (833 km northeast of Perth, Fig 1A).

The Sturt Meadows calcrete hosts a very shallow aquifer, located two to four metres below the surface, and accessible through bores characterised by water depths ranging from a few centimetres to ten metres. The bore grid was initially drilled for mineral exploration and comprises 115 bore holes of between 5–11 m in depth forming a 1.4 km by 2.5 km (3.5 km$^2$) area (Fig 1A). The bores are unlined, except for about the upper 0.5 m which are lined with 10 cm diameter PVC pipe for stability, and capped [36]. Three sampling campaigns–two of them corresponding to low rainfall periods (LR) and one during the wet season (high rainfall, HR) [37]–were carried out in July and November 2017, and March 2018. More details about the sampling design, monitoring of water depth and hydrogeological background at Sturt Meadows can be found in Saccò et al. [38].

The high morphologically (Fig 1B) and taxonomically diverse stygofaunal community at Sturt Meadows comprises three sister species of subterranean beetles (*Paroster macrosturtensis* (Watts & Humphreys 2006), *Paroster mesosturtensis* (Watts & Humphreys 2006) and *Paroster microsturtensis* (Watts & Humphreys 2006) and respective larvae)), three species of amphipods (*Scutachiltonia axfordi* (King, 2012), *Yilgarniella sturtensis* (King, 2012) and *Stygochiltonia bradfordae* (King, 2012)), aquatic worms (family Tubificidae (Vejdovský, 1884)) and water mites (order Oribatida; Dugès, 1834). Within the stygobiotic meiofaunal community, two species of harpacticoids (*Novanitocrella cf. aboriginesi* (Karanovic, 2004), *Schizopera cf. austin-downsi* (Karanovic, 2004) and four species of cyclopoids: *Halicyclops kieferi* (Karanovic, 2004), *Halicyclops cf. ambiguous* (Kiefer, 1967), *Schizopera slenderfurca* (Karanovic & Cooper, 2012) and *Fierscyclops fiersi* (De Laurentiis et al., 2001)) can be found.

Adult and larval stygofaunal specimens were collected by hauling a small weighted plankton net (mesh 100 μm, [36]) five times from the bottom through the water column of 30 boreholes (Fig 1A) selected by simple random sampling [38]. Stygofaunal abundance data across the boreholes are reported in S1 Table.

All biological samples were kept frozen (−20˚C) in darkness until further processing in the laboratory where individual organisms were counted and identified (and consequently separated) to the lowest taxonomic level *via* optical microscopy and reference to specific taxonomic keys. Roots and sediment samples from the bottom of the aquifer were obtained through the stygofaunal haul netting procedure, and were separated by using sterile glass pipettes during the sorting in the laboratory according to the sampling campaign (LR or HR). Sediment

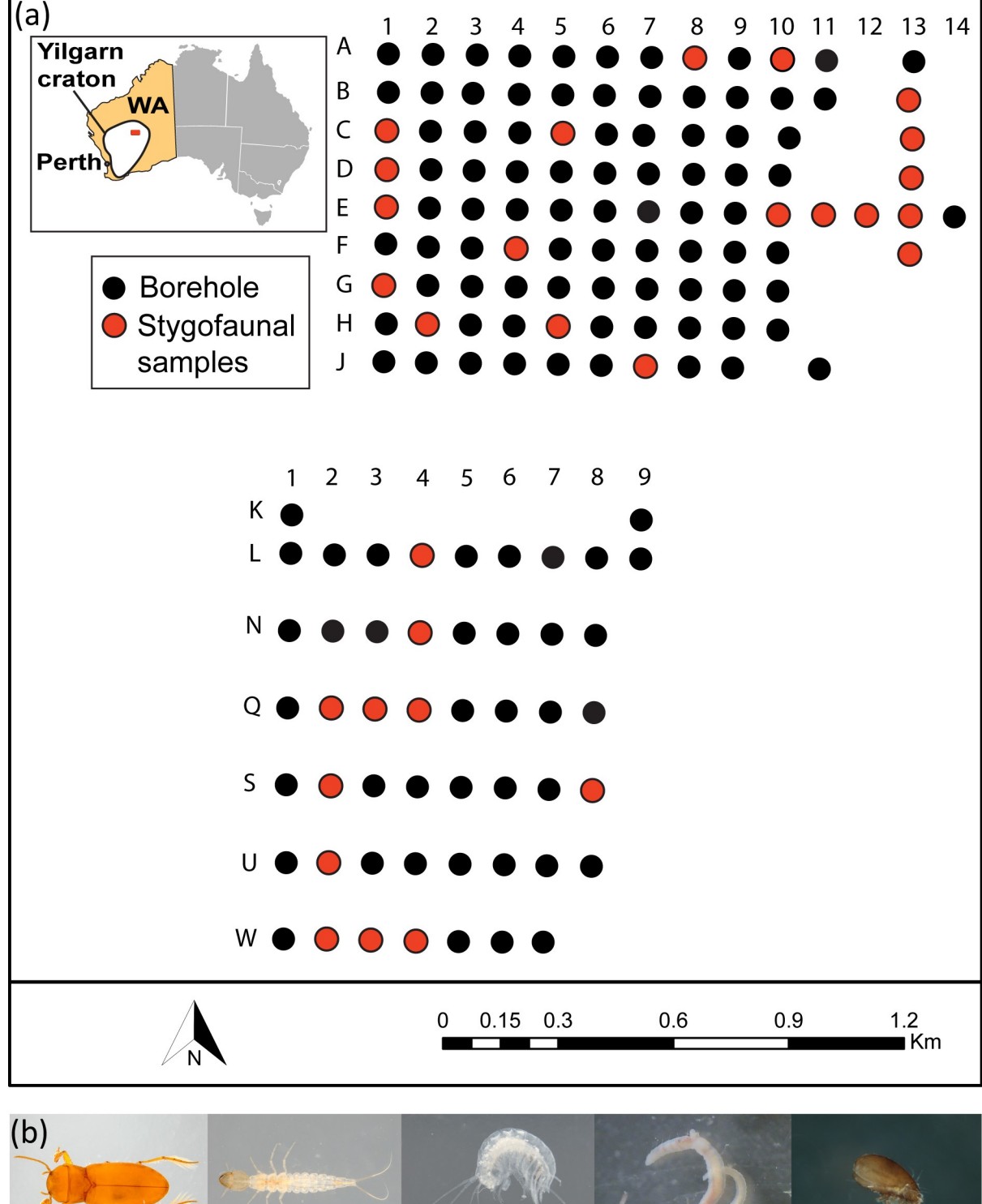

**Fig 1.** a) Borehole grid and its location in the Yilgarn region, Western Australia. b) Photos of some specimens from the bore samples (from left to right *Paroster macrosurtensis* adult, *Paroster microsturtensis* larvae, *Scutachiltoni axfordi*, Oligochaeta sp. and Oribatida sp.).

samples were soaked in acid (0.1 N HCl) to remove inorganic carbon and dried at 60˚C for 24 hours. Given the delicacy of the hydrological dynamics in shallow calcretes [39], extensive water extractions spread along the bores were avoided and preliminary tests were carried out to quantify the potential risk of dewatering the calcrete.

Bores D13 and W4 host groundwater systems which are representative of the geological conformations of the area—phreatic and vadose calcretes interspersed with clay material—and were finally selected because of their hydrological and biotic stability (lowest risk of drying and representative ranges of Sturt Meadows stygofaunal diversity) [38]. Water samples for POC (particulate organic carbon) analysis were collected using a submersible centrifugal pump (GEOSub 12V Purging Pump) after wells were purged of three well-volumes and stabilisation of in-field parameters was observed. POC samples were obtained by filtering water from the bores D13 and W4 through GF/F filters (pre-combusted for 12 hours at 450˚C), washed with 1.2 N HCl to remove any inorganic carbon, and subsequently dried at 60˚C for 24 hours. The field site was accessed and samples were collected with permit approval (permit number 08-003150-1) from the Department of Parks and Wildlife of Western Australia.

### Sample preparation and study design

All individuals from a single taxon were pooled for each sampling campaign (LR1, LR2 or HR) and subsequently washed with MilliQ water to remove external contaminants. Subsequently, samples were oven dried at 60˚C overnight and crushed to a fine powder which was stored at −20˚C until further analysis (Fig 2).

Due to sample size constraints, the samples for each taxon from the two low rainfall periods were further combined. Previous metagenomics investigations, together with mesocosm experiments and field observations at Sturt Meadows provided some information about the trophic habits of beetles and amphipods [40]. Adult subterranean beetles had active predatory feeding on epigean amphipods and copepods (including group feeder behaviours) together with scavenger habits (and potential active predatory pressures) on sister species. Beetle larvae (third and last instar) showed opportunistic predatory habits with a range of prey from copepods and amphipods to adult beetles from the three species (inter and intraspecific cannibalism), while amphipods displayed predation of copepods, epilithic biofilm grazing, root shredding and sediment filter feeding.

### Bulk stable isotope analysis

$\delta^{13}C$ and $\delta^{15}N$ SIA on bulk homogenised samples of sediment, roots and stygofauna (respectively 1.28 mg, 0.08–0.14 mg and 0.63–2.79 mg per samples, S2 Table) were performed at the Australian Nuclear Science and Technology Organisation (ANSTO, Sydney). Samples were loaded into tin capsules and analysed with a continuous flow isotope ratio mass spectrometer (CF-IRMS, Delta V Plus, Thermo Scientific Corporation, U.S.A.), interfaced with an elemental analyser (Thermo Fisher Flash 2000 HT EA, Thermo Electron Corporation, U.S.A.) following the procedure of Mazumder et al. [41]. $\delta^{13}C$ values are reported in per mil (‰) relative to the Vienna Peedee Belemnite (VPDB), while $\delta^{15}N$ values are reported relative to reference $N_2$ of known nitrogen isotopic composition (in ‰), previously calibrated against the AIR international isotope standard. $\delta^{13}C$ POC (0.6 mg, S2 Table) was analysed at the Western Australian Biogeochemistry Centre at The University of Western Australia using a GasBench II coupled with a Delta XL Mass Spectrometer (Thermo-Fisher Scientific). Results have a precision of ± 0.10 ‰, and are reported relative to the NBS19 and NSB18 international carbonate standard [42].

## Single amino acid carbon and nitrogen isotope analysis

**δ¹³C CSIA.**   Stygofaunal samples (0.16–2.89 mg per sample, S2 Table) were hydrolysed under vacuum with 0.5 to 1 ml of amino acid-free 6 M HCl (Sigma-Aldrich) at 110 ˚C for 24 h. The protein hydrolysates were dried overnight in a rotary vacuum concentrator and stored in a freezer. Prior to analysis, the samples were resolved in Milli-Q water and 10 µl of 1-mmol solution of 2-aminoisobutyric acid (Sigma-Aldrich) was added as internal standard. The sample stock had a concentration of approximately 8 to 10 mg/ml, which was further diluted as

1) SORTING UNDER STEREOMICROSCOPE

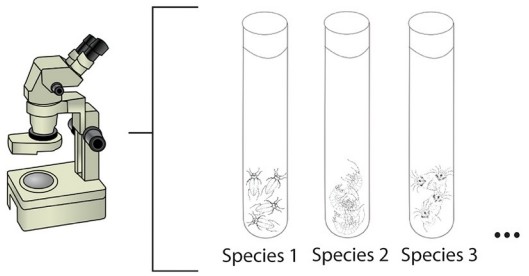

2) PRETREATMENT IN THE LAB

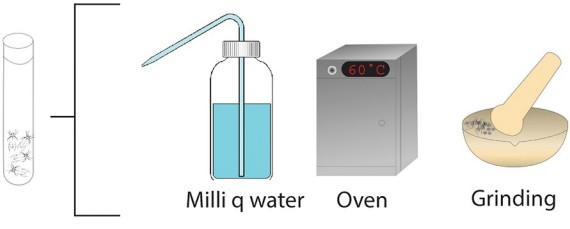

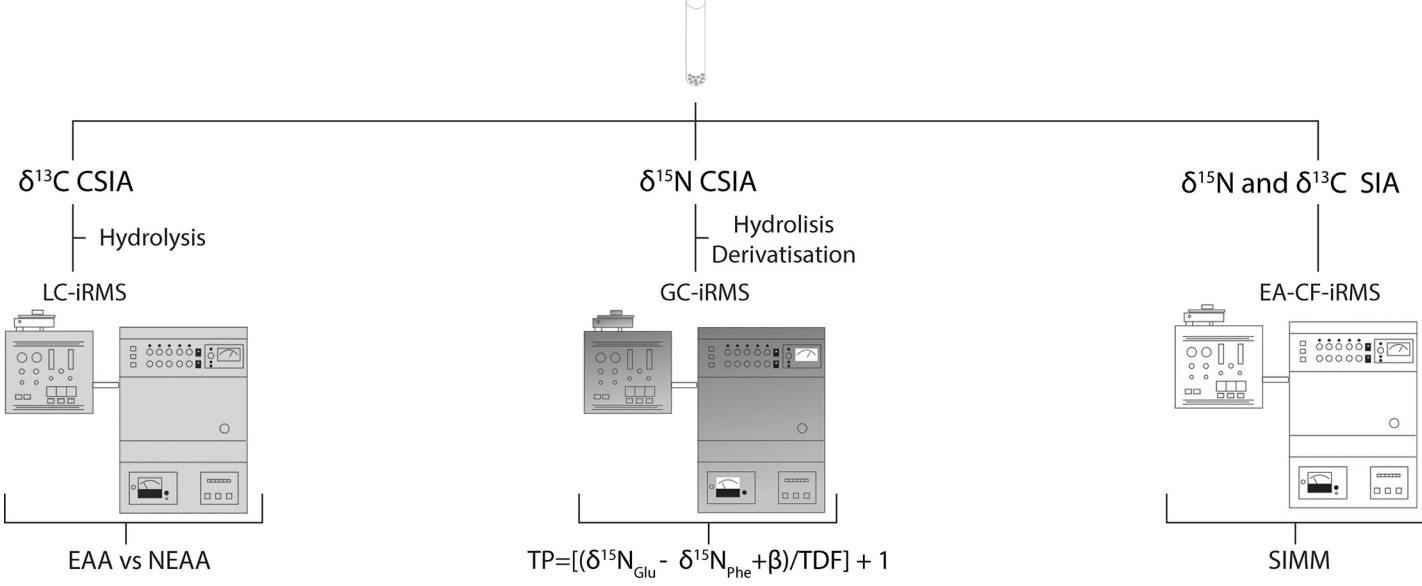

**Fig 2. Methodological scheme of the study for stygofaunal samples (including copepods for bulk SIA).** EAA: essential amino acids; NEAA: non-essential amino acids; TP: trophic position; TDF: trophic discrimination Factor; β = ratio between δ15NGlu and δ15NPhe values in primary producers; SIMM: stable isotopes mixing models; LC-iRMS: Liquid Chromatography-isotope Ratio Mass Spectrometry; GC-iRMS: Gas Chromatography-isotope Ratio Mass Spectrometry; EA-CF-iRMS: Elemental Analyser-Continuous Flow-isotope Ratio Mass Spectrometry.

needed. Single amino acid carbon isotope analysis was carried out at the La Trobe Institute for Molecular Sciences (LIMS, La Trobe University, Melbourne, Australia) using an Accela 600 pump connected to a Delta V Plus Isotope Ratio Mass Spectrometer *via* a Thermo Scientific LC Isolink (Thermo Scientific).

The amino acids were separated using a mixed mode (reverse phase/ion exchange) Prime-sep A column (2.1 x 250 mm, 100˚C, 5 μm, SIELC Technologies) following the chromatographic method described in Mora et al. [43], after Smith et al. [44]. Mobile phases are those described in Mora et al. [45]. Percentage of Phases B and C in the conditioning run, as well as flow rate of the analytical run and timing of onset of 100% Phase C were adjusted as needed. Samples were injected onto the column in the 15 μl—partial loop or no waste—injection mode, and measured in duplicate or triplicate.

**δ$^{15}$N CSIA.** CSIA nitrogen analyses were undertaken at the Organic Geochemistry Unit of the University of Bristol, UK. To extract the AAs, crushed samples (2.47–5.19 mg per sample, S2 Table) were hydrolysed in culture tubes (6 M HCl, 2 ml, 100˚C, 24 h). A known quantity of norleucine (1 mg mL–1 in 0.1 M HCl) was added to each sample as an internal standard prior to hydrolysis. After heating, the tubes were allowed to cool then after centrifugation (3000 rpm, 5 min) the supernatant containing the hydrolysate from each tube was transferred to a clean culture tube and dried under $N_2$ whilst being heated to 70˚C. Once dry, each sample was re-dissolved in 0.1 M HCl and stored in the dark at -18˚C until required for analysis.

The derivatisation procedure followed Styring et al. [46] and included isopropylation, with a 4:1 mixture of 2-propanol and acetyl chloride heating to 100˚C for 1 hour, the reaction was quenched by rapidly cooling in a freezer. After removing the residual solvents under $N_2$, acetylation of the amino group was achieved by adding a 5:2:1 mixture of acetone, triethylamine and acetic anhydride then heating to 60˚C for 10 minutes before being allowed to cool. The derivatised AAs were isolated *via* liquid-liquid separation, residual solvent being removed by evaporating under $N_2$. Samples were again stored at -18˚C until required for analysis.

A Thermo Finnigan Delta Plus XP isotope ratio mass spectrometer (Thermo Scientific, Bremen, Germany) was used to determine the δ$^{15}$N values of derivatised AAs. The mass spectrometer (EI, 100 eV, three Faraday cup collectors for m/z 28, 29 and 30) was interfaced to a Trace 2000 gas chromatograph *via* a Combustion III interface (CuO/NiO/Pt oxidation reactor maintained at 980˚C and reduction reactor of Cu wire maintained at 650˚C), both from Thermo Scientific.

Samples were dissolved in ethyl acetate and 1μl of solution was injected *via* a PTV injector. Helium at a flow of 1.4 ml min$^{-1}$ was used as the carrier gas and the mass spectrometer source pressure was maintained at 9 X 10–4 Pa. The separation of the AAs was accomplished using a DB-35 capillary column (30 m X 0.32 mm i.d., 0.5 mm film thickness; Agilent Technologies, Winnersh, UK). The oven temperature of the GC started at 40˚C where it was held for 5 min before heating at 15˚C min$^{-1}$ to 120˚C, at 3˚C min$^{-1}$ to 180˚C, at 1.5˚C min$^{-1}$ to 210˚C and finally at 5˚C min$^{-1}$ to 270˚C and held for 1 min. A Nafion dryer removed water and a cryogenic trap was employed to remove $CO_2$ from the oxidised and reduced analyte.

All the δ$^{15}$N values are reported relative to reference $N_2$ of known nitrogen isotopic composition, previously calibrated against the AIR international isotope standard, introduced directly into the ion source in four pulses at the beginning and end of each run. Each reported value is a mean of duplicate δ$^{15}$N determinations. A standard mixture of derivatised AAs of known δ$^{15}$N values was analysed every three runs in order to monitor instrument performance.

## Data treatment and statistical analysis

Only AAs that returned results for each taxon were considered. EAA and NEAA were separated according to the classification provided by Boudko [47]. EAAs were used in the

interpretation of carbon flows—and potential shifts in OM incorporations—because they persist through the trophic chain [48] due to the little fractionation they undergo when incorporated into consumer's tissue [49]. NEAA, which are subjected to much greater fractionation because of their *de novo* biosynthesis mainly from intermediates of the Krebs cycle (serine (Ser), glycine (Gly) and alanine (Ala)) and glycolysis (glutamic acid (Glx), aspartic acid (Asx) and proline (Pro)) [50], were compared to EAA to investigate taxa-specific carbon isotopic trends (biosynthesis vs assimilation through diet) across the two rainfall periods (LR and HR).

All the statistical analyses were performed in R software version 3.6.0 (Development-Core-Team, 2016). Analysis of variance (ANOVA) coupled with Tukey's HSD pairwise comparisons (R-package 'stats') were employed to inspect significant differences between $\delta^{13}C_{EAA}$ (Val, Phe and Arg) and $\delta^{13}C_{NEAA}$ (Krebs (Ser, Gly and Ala) and glycolysis (Asx, Glx and Pro) cycles) within the different rainfall conditions (LR and HR). Principal component analyses (PCA, R-package 'vegan') and Linear Discriminant Analysis (LDA, R-package 'vegan') among EAA was performed to explore sample distribution in the multi-dimensional space. Determination of EAA driving sample variability in the PCA was carried out *via* function *fviz_contrib* (R-package 'factoextra').

Trophic positions (TP) were calculated using the methodology reported by Chikaraishi et al. [33]:

$$TP = [(\delta^{15}N_{Glu} - \delta^{15}N_{Phe} + \beta)/TDF] + 1$$

where $\delta^{15}N_{Glu} = \delta^{15}N$ of glutamic acid, $\delta^{15}N_{Phe} = \delta^{15}N$ of phenylalanine, $\beta$ = ratio between $\delta^{15}N_{Glu}$ and $\delta^{15}N_{Phe}$ values in primary producers, and TDF = the trophic discrimination factor at each shift of trophic position.

Incorporation of source carbon from terrestrial vegetation has previously been reported at Sturt Meadows, with roots from surficial saltbush vegetation (C3 metabolism) frequently found in the groundwater [40]. $\beta$ was accordingly assigned the value of +8.4 ± 1.6 ‰, which is the established value for aquatic food webs involving C3 plants [31]. Although other carbon sources are possible in groundwaters, as they are not established in this system, a conservative approach has been taken in using the value of an evidenced source. TDF was assigned the value of 7.6 ± 1.2‰, based on Steffan et al. [51] who showed it did not vary across trophic levels one to four in multiple controlled-feeding experiments, and for trophic levels one to five in a natural food chain, using terrestrial arthropod species [28].

Pairwise comparisons for $\delta^{15}N$ were carried out with the same approach as for the carbon CSIA data. Robustness and consistency between CSIA and SIA data from beetles and amphipods were inspected using Pearson correlations (function *rcorr* in R-package 'Hmisc'). SIMM (Stable Isotope Mixing Models, R-package 'simmr') were then applied to establish dietary proportions of the key ecological taxa (Fig 2). Since a specific trophic discrimination factor has not been calculated for stygofauna, we used the widely accepted values of 3.4 ± 2 ‰ for nitrogen and 0.5 ± 1 ‰ for carbon [52]. Markov chain Monte Carlo (MCMC) algorithms were used for simulating posterior distributions in SIMM, and MCMC convergence was evaluated using the Gelman-Rubin diagnostic by using 1.1 as a threshold value for analysis validation.

## Results

### Stygofaunal carbon fluxes

During LR, $\delta^{13}C$ average values of AAs ($\delta^{13}C_{NEAA[LR]}$ and $\delta^{13}C_{EAA[LR]}$) spanned from -31.52 ‰ (Phe) to -5.72 ‰ (Gly). Similar values were found under HR conditions ($\delta^{13}C_{NEAA[HR]}$ and $\delta^{13}C_{EAA[HR]}$), ranging from -31.55 ‰ (Phe) to -4.92 ‰ (Ser) (Table 1).

**Table 1. Low (LR) and high rainfall (HR) carbon amino acids spectrum ($\delta^{13}$C values) for stygofaunal specimens separated by non-essentials (NEAA: aspartic acid (Asx), serine (Ser), glutamic acid (Glx), glycine (Gly), alanine (ala), and proline (Pro)), and essentials (EAA: valine (Val), phenylalanine (Phe), and arginine (Arg)).** Average values (and standard deviation) for the analytical replicates are shown. *P* values for ANOVA Tukey's HSD pairwise comparisons between NEAA and EAA are also illustrated.

| Taxon | ID | NEAA | | | | | | EAA | | | NEAA[2] vs EAA[3] | NEAA[4] vs EAA[3] |
|---|---|---|---|---|---|---|---|---|---|---|---|---|
| | | Asx | Ser | Glx | Gly | Ala | Pro | Val | Phe | Arg | Krebs cycle | Glycolysis |
| **LR** | | | | | | | | | | | | |
| *Paroster macrosturtensis* | B | -17.64 ±0.59 | -10.43 ±0.1 | -18.74 ±0.53 | -10.01 ±0.12 | -20.01 ±0.12 | -17.85 ±0.53 | -24.39 ±0.55 | -24.44 ±0.13 | -19.56 ±0.55 | P < 0.05 | P < 0.05 |
| *Paroster mesosturtensis* | M | -18.85 ±0.6 | -11.32 ±0.16 | -20.3[1] | -12.22 ±0.6 | -21.23 ±0.52 | -15.10 ±0.63 | -24.55 ±0.27 | -26.24 ±0.23 | -20.42 ±0.07 | P < 0.05 | P < 0.05 |
| *Paroster microsturtensis* | S | -22.42 ±0.18 | -12.39 ±0.62 | -22.42[1] | -14.84 ±0.64 | -24.24 ±0.35 | -18.67 ±0.62 | -27.73 ±0.64 | -28.98 ±0.07 | -23.19 ±0.54 | P < 0.05 | P < 0.05 |
| *Paroster macrosturtensis* larvae | Blv | -20.56 ±0.11 | -9.22 ±0.18 | -20.20[1] | -14.17 ±0.6 | -22.01 ±0.58 | -19.31 ±0.6 | -25.97 ±0.17 | -26.48 ±0.12 | -20.59 ±0.24 | P < 0.05 | P = 0.0618 |
| *Paroster mesosturtensis* larvae | Mlv | -19.09 ±0.04 | -6.70 ±0.18 | -20.05 ±0.43 | -16.12 ±0.65 | -21.12 ±0.37 | -16.38 ±0.3 | -24.92 ±0.09 | -27.08 ±0.1 | -19.59 ±0.3 | P < 0.05 | P < 0.05 |
| *Paroster microsturtensis* larvae | Slv | -19.1 ±0.5 | -8.41 ±0.05 | -17.75 ±0.03 | -14.20 ±0.63 | -21.57 ±0.15 | -20.38 ±0.2 | -25.34 ±0.07 | -27.02 ±0.38 | -18.59 ±0.21 | P < 0.05 | P < 0.05 |
| *Scutachiltonia axfordi* | AM1 | -16.1 ±1.3 | -5.87 ±0.8 | -14.09 ±1.4 | -5.72 ±1.01 | -20.42 ±2.88 | -16.68 ±0.2 | -24.55 ±2.01 | -21.94 ±0.28 | -15.31 ±0.74 | P < 0.05 | P < 0.05 |
| *Yilgarniella sturtensis* | AM2 | -19.09 ±0.08 | -8.31 ±0.41 | -21.35 ±3.77 | -6.81 ±0.42 | -20.29 ±0.09 | -16.63 ±0.12 | -25.48 ±0.17 | -25.6 ±1.29 | -19.56 ±1.39 | P < 0.05 | P = 0.0511 |
| *Stygochiltonia bradfordae* | AM3 | -21.7 ±0.1 | -9.15 ±0.64 | -24.65 ±4.38 | -9.00 ±0.33 | -24.08 ±0.19 | -22.76 ±0.64 | -28.84 ±0.25 | -28.27 ±0.1 | -23.54 ±0.38 | P < 0.05 | P = 0.0952 |
| Tubificidae sp. | OL | -21.7 ±0.24 | -16.01 ±0.44 | -24.33 ±0.31 | -20.93[1] | -26.36±0.1 | -25.78 ±0.36 | -31.46 ±0.1 | -31.52 ±0.23 | -27.58 ±0.04 | P < 0.05 | *P < 0.05* |
| Oribatida sp. | OR | -20.44 ±0.63 | -11.99 ±0.23 | -17.77[1] | -14.12 ±0.47 | -21.18 ±0.07 | -19.31 ±0.61 | -26.36 ±0.03 | -24.33 ±0.55 | -18.85 ±0.2 | P = 0.0955 | P = 0.0826 |
| **HR** | | | | | | | | | | | | |
| Paroster macrosturtensis | B | -18.67 ±0.45 | -11.62 ±0.29 | -18.19 ±0.59 | -11.31 ±0.11 | -19.648 ±0.45 | -16.83 ±0.56 | -25.44 ±0.64 | -26.3 ±0.61 | -20.40 ±0.6 | P < 0.05 | P < 0.005 |
| Paroster mesosturtensis | M | -23.88[1] | -18.06[1] | -23.56[1] | -16.64[1] | -25.96[1] | -21.99[1] | -29.768[1] | -31.08[1] | -26.05[1] | P < 0.05 | P < 0.05 |
| Paroster microsturtensis | S | -20.87[1] | -11.46[1] | -20.8[1] | -13.94[1] | -22.62[1] | -17.82[1] | -26.773[1] | -29.04[1] | -22.63[1] | P < 0.05 | P < 0.005 |
| Paroster macrosturtensis larvae | Blv | -21.64 ±0.55 | -12.58[1] | -20.63[1] | -13.56[1] | -24.22±0.1 | -19.43[1] | -28.12 ±0.51 | -28.22 ±0.57 | -22.35 ±0.58 | P < 0.05 | P < 0.05 |
| Paroster mesosturtensis larvae | Mlv | -23.88[1] | -18.06[1] | -23.56[1] | -16.94 ±0.42 | -25.97[1] | -21.99[1] | -29.38 ±0.55 | -31.08[1] | -26.05[1] | P < 0.05 | P < 0.05 |
| Paroster microsturtensis larvae | Slv | -24.15 ±0.35 | -18.03 ±0.59 | -24.24 ±0.01 | -18.74[1] | -26.11 ±0.11 | -22.19[1] | -30.074[1] | -31.55[1] | -26.59 ±0.2 | P < 0.05 | P < 0.005 |
| Scutachiltonia axfordi | AM1 | -23.73 ±0.6 | -12.36 ±0.21 | -23.55 ±0.35 | -12.89 ±0.26 | -24.13 ±0.47 | -21.68 ±0.46 | -29.52 ±0.52 | -29.56 ±0.43 | -23.80 ±0.54 | P < 0.05 | P < 0.05 |
| Yilgarniella sturtensis | AM2 | -23±0.01 | -13.67 ±0.25 | -23.60 ±0.42 | -14.92 ±0.32 | -26.28 ±0.65 | -24.03 ±0.45 | -30.87 ±0.13 | -30.31 ±0.63 | -24.99 ±0.58 | P < 0.05 | P < 0.05 |
| Stygochiltonia bradfordae | AM3 | -23.39 ±0.2 | -12.37 ±0.62 | -22.68 ±0.01 | -13.19 ±0.01 | -23.8465 ±0.14 | -21.41 ±0.51 | -28.7 ±0.22 | -28.28 ±0.52 | -22.94 ±0.11 | P < 0.05 | P < 0.05 |
| Tubificidae sp. | OL | -20.25[1] | -14.11[1] | -20.25[1] | -15.6[1] | -23.94[1] | -20.47[1] | -27.93[1] | -28.79[1] | -21.62[1] | P < 0.05 | P = 0.0951 |
| Oribatida sp. | OR | -20.42 ±0.64 | -4.92[1] | -22.11 ±0.61 | -5.9 ±0.11 | -22.19 ±0.09 | -19.69 ±0.01 | -27.52 ±0.11 | -27.41[1] | -21.02 ±0.39 | P < 0.005 | P < 0.05 |

[1]Unique run

[2]Calculated as average value of Ser, Gly and Ala

[3]Calculated as average value of Val, Phe and Arg

[4]Calculated as average value of Asx, Glx and Pro

$\delta^{13}C_{NEAA[LR]}$ average values varied from -26.36 ‰ (Ala) to -5.72 ‰ (Gly), similar values to $\delta^{13}C_{NEAA[HR]}$ spanning from -26.28 ‰ (Ala) to -4.92 ‰ (Ser). Overall, $\delta^{13}C_{EAA}$ showed trends towards more negative values than $\delta^{13}C_{NEAA}$, which is involved in both Krebs and glycolytic cycles (ANOVA, P <0.005). This is consistent with the enrichment of NEAA during biosynthesis in the organism. With the exception of water mites (OR), *P. macrosturtensis* larvae (Blv), *Y. sturtensis* (AM2), *S. bradfordae* (AM3) under LR, and oligochaetes (OL) under HR, pairwise comparisons between $\delta^{13}C_{EAA}$ and $\delta^{13}C_{NEAA}$ confirmed a shift towards more negative values across the stygofaunal community (Table 1).

Neither PCAs nor LDAs on EAA distinguished different clusters within taxa or main groups (adult and larval beetles, and amphipods) nor among different rainfall periods (LR and HR). All three EAA correlated positively and significantly ($P < 0.005$), with phenylalanine and valine being the most informative AAs explaining the isotopic variability across stygofauna (~70%). $\delta^{13}C$ values of valine ($\delta^{13}C_{Val}$) and phenylalanine ($\delta^{13}C_{Phe}$) show that, with the exception of *P. microsturtensis* (S) and *S. bradfordae* (AM3), the entire stygofaunal community experienced a significant shift towards more $^{13}C$-depleted values under HR (MANOVA, $P < 0.005$) (Fig 3, Table 2).

Within the significant trends, *P. macrosturtensis* adults and larvae (B and Blv) showed the smallest change in carbon values (B: $\delta^{13}C_{Val+Phe}$ = -2.91; Blv: $\delta^{13}C_{Val+Phe}$ = -3.89) between rainfall regimes, while amphipods *S. axfordi* and *Y. sturtensis* (AM1 and AM2) showed the largest depletion (AM1: $\delta^{13}C_{Val+Phe}$ = -12.59 ‰; AM2: $\delta^{13}C_{Val+Phe}$ = -10.10 ‰), suggesting differential carbon incorporations under HR conditions.

## $\delta^{15}N$ and trophic levels

$\delta^{15}N_{Glu}$ average values varied between 15.4±0.4‰ (AM3[HR]) and 22.31±0.29‰ (M[HR]), while $\delta^{15}N_{Phe}$ values ranged from 10.67±0.45‰ (AM3[HR]) to 14.53±0.06‰ (M[HR]). When converted to trophic positions, the stygofaunal community at Sturt Meadows shows a truncated trophic chain, clustering around the secondary consumer level (Fig 4).

Under LR conditions, *P. macrosturtensis* larvae (Blv) show the highest trophic position (TP = 3.33±0.02), while water mites (OR) sit at the lowest (2.78±0.09). Under HR conditions, *P. microsturtensis* adults (S) have the highest trophic position (3.27±0.01), whilst *S. bradfordae* (AM3) show the lowest value (TP = 2.73±0.01). Due to the low abundances it wasn't possible to analyse biochemical fingerprints from water mites (OR[HR]: 37 individuals) and *P. microsturtensis* larvae (Slv[HR]: 10 individuals) during the wet season (HR) (S1 Table).

Overall, adult beetles (B, M and S) revealed higher trophic levels (TP>3) than amphipods (AM1, AM2 and AM3, TP<3). However, B, M and S did not show statistically higher values than AM1 under LR, the same pattern seen in *P. mesosturtensis* (M) under HR.

*S. bradfordae* (AM3) and *P. macrosturtensis* larvae (Blv) are the only organisms to show a statistically significant change in their TP values between LR and HR (Table 3), with both decreasing trends.

## Food web dynamics

CSIA-based TP correlated significantly with SIA $\delta^{15}N$ and $\delta^{13}C$ values both under LR (P<0.05 in both cases) and HR conditions (P<0.01 and P<0.05 respectively). Under the latter conditions, $\delta^{13}C_{Val}$ values correlated significantly with CSIA-based TP (P<0.05). Copepods are generally thought to sit at the base of the food web [53,54]. However, these were analysed only via bulk SIA due to organism and sample size, and so could not be included in the TP analysis. They showed more $^{13}C$-depleted $\delta^{13}C$ (cyclopoids: $\delta^{13}C_{LR}$ = -20.5‰, $\delta^{13}C_{HR}$ = -21.9‰; harpacticoids: $\delta^{13}C_{LR}$ = -20.6‰, $\delta^{13}C_{HR}$ = -23.5‰) and enriched $\delta^{15}N$ (cyclopoids: $\delta^{15}N_{LR}$ =

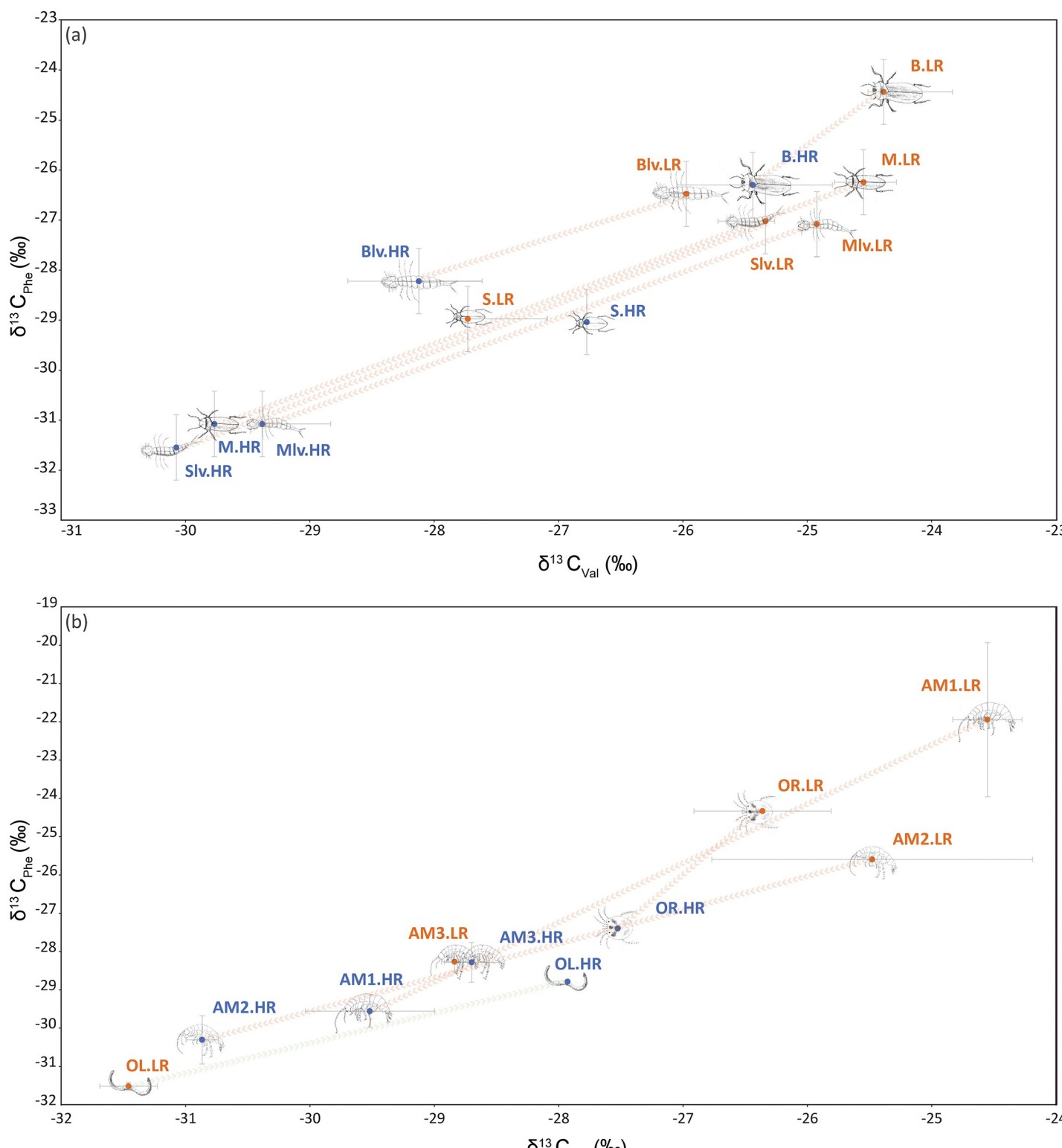

**Fig 3.** Biplot of $\delta^{13}C_{Phe}$ values vs. $\delta^{13}C_{Val}$ values for a) beetles (B, M, S, Blv, Mlv and Slv) and b) amphipods (AM1, AM2 and AM3), water mites (OR) and aquatic worms (OL). Red arrows indicate significant decreasing trends between LR and HR, while green arrows indicate increasing trends within rainfall periods. Refer to Table 1 for taxa IDs.

**Table 2. Tuckey's post hoc pairwise comparisons between phenylalanine and valine values under low (LR) and high (HR) rainfall conditions.** In bold significant results.

| Taxon | ID | Phe | | | Val | | |
|---|---|---|---|---|---|---|---|
| | | d.f. | T-ratio | P | d.f. | T-ratio | P |
| *Paroster macrosturtensis* | B | 28 | -4.497 | **< .0005** | 28 | -2.163 | **< .005** |
| *Paroster mesosturtensis* | M | 28 | -4.846 | **< .0001** | 28 | -3.297 | **< .005** |
| *Paroster microsturtensis* | S | 28 | -0.149 | 0.8829 | 28 | 1.967 | 0.0592 |
| *Paroster macrosturtensis* larvae | Blv | 28 | -4.218 | **< .0005** | 28 | -4.42 | **< .0005** |
| *Paroster mesosturtensis* larvae | Mlv | 28 | -9.657 | **< .0001** | 28 | -9.16 | **< .0001** |
| *Paroster microsturtensis* larvae | Slv | 28 | -10.933 | **<0.001** | 28 | -9.73 | **< .0001** |
| *Scutachiltonia axfordi* | AM1 | 28 | -18.4 | **< .0001** | 28 | -10.2 | **< .0001** |
| *Yilgarniella sturtensis* | AM2 | 28 | -11.383 | **< .0001** | 28 | -11.067 | **< .0001** |
| *Stygochiltonia bradfordae* | AM3 | 28 | -0.037 | 0.9704 | 28 | 0.282 | 0.7797 |
| Tubificidae sp. | OR | 28 | -7.418 | **< .0001** | 28 | -2.389 | **<0.05** |
| Oribatida sp. | OL | 28 | 6.594 | **< .0001** | 28 | 7.252 | **< .0001** |

13.9‰, $\delta^{15}N_{HR}$ = 14.5‰; harpacticoids: $\delta^{15}N_{LR}$ = 11.9‰, $\delta^{15}N_{HR}$ = 15.8‰) values under HR (Fig 5A and 5B), indicating that the change in rainfall regimes could play a role in stygobiotic meiofaunal biochemical incorporations.

Amphipods AM1 and AM2 sat at the base of the trophic web under both rainfall conditions (TPs always below three, Table 3), and SIA carbon values ($\delta^{13}C$) confirmed a shift–already pinpointed *via* CSIA—towards more $^{13}C$-depleted carbon sources under HR. AM3, the smallest and rarest amphipod species in the calcrete, did not allow bulk SIA analyses due to the low abundances detected (LR (average value between LR1 and LR2): 27; HR: 19, S1 Table).

With respect to dietary preferences, for the amphipod *S. axfordi* (AM1), mixing models suggest that roots (and hosted microbial flora) contributed the greatest proportion (50%) during low rainfall conditions (LR) (Fig 6). The remaining diet was composed of POC (16.1%, derived from allochthonous carbon incorporations, a potential organic source for microbes), copepods (harpacticoids (13.9%) and cyclopoids (11.9%)) and sediment (8.1%) (i.e. OM laying at the bottom of the aquifer or epilithic biofilms).

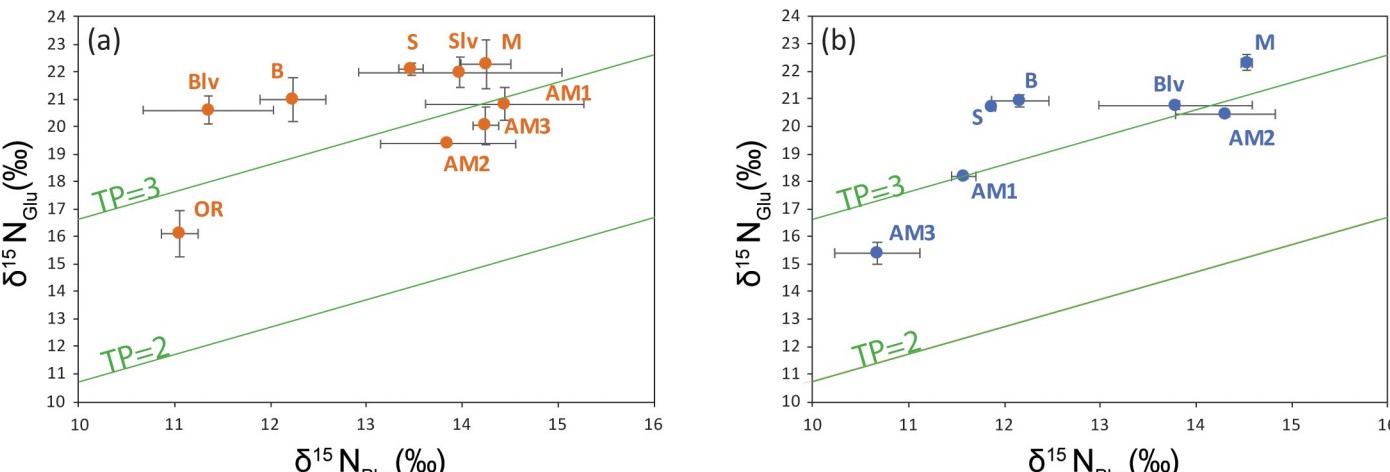

**Fig 4.** Calculated trophic positions (TP) of the stygofaunal specimens studied under LR (a) and HR (b) conditions.

**Table 3. δ¹⁵N_Glu, δ¹⁵N_Phe and TP values (±SD) during LR and HR regimes.** Pairwise comparisons within taxa from the same rainfall conditions and between rainfall periods (in bold significant patterns) for the same taxa are also illustrated. Taxa sharing the same letter do not differ significantly (Tukey's HSD test, P < 0.05).

| | $\delta^{15}N_{Glu}$ (‰) | | $\delta^{15}N_{Phe}$ (‰) | | TP | | TP pairwise comparison | | |
| --- | --- | --- | --- | --- | --- | --- | --- | --- | --- |
| | LR | HR | LR | HR | LR | HR | LR | HR | LR vs HR |
| B | 20.99±0.79 | 20.93±0.23 | 12.23±0.34 | 12.16±0.30 | 3.26±0.06 | 3.26±0.01 | de | e | 0.9712 |
| M | 22.29±0.89 | 22.31±0.29 | 14.25±0.26 | 14.53±0.06 | 3.17±0.08 | 3.13±0.03 | bcde | cde | 0.5415 |
| S | 22.12±0.23 | 20.69±0.08 | 13.47±0.13 | 11.87±0.03 | 3.25±0.01 | 3.27±0.01 | de | e | 0.6698 |
| Blv | 20.61±0.5 | 20.77±0.14 | 11.35±0.68 | 13.79±0.80 | 3.33±0.02 | 3.03±0.09 | e | bcd | **< .0001** |
| Slv | 21.99±0.55 | Na | 13.98±1.06 | Na | 3.16±0.21 | Na | bcde | Na | Na |
| AM1 | 20.84±0.62 | 18.19±0.1 | 14.44±0.83 | 11.57±0.13 | 2.95±0.03 | 2.98±0.01 | abcd | bc | 0.6193 |
| AM2 | 19.38±0.01 | 20.45±0.08 | 13.85±0.7 | 14.31±0.52 | 2.84±0.09 | 2.92±0.08 | abc | b | 0.135 |
| AM3 | 20.04±0.7 | 15.4±0.4 | 14.24±0.13 | 10.67±0.45 | 2.87±0.07 | 2.73±0.01 | ab | a | **< .05** |
| OR | 16.11±0.85 | Na | 11.05±0.19 | Na | 2.78±0.09 | Na | a | Na | Na |

Under HR conditions, the POC dietary contribution reached 66.1%, while roots plummeted to 3.3% (Fig 6). Overall, amphipod *Y. sturtensis* (AM2) showed the same dietary patterns as AM1.

Adult beetles *P. macrosturtensis* (B) and *P. mesosturtensis* (M) show only slight depletions in their isotopic values during HR in bulk δ¹³C and δ¹⁵N SIA, in contrast to the larger changes seen in the CSIA data. *P. microsturtensis* (S), which showed an isotopic enrichment in CSIA, counter to the rest of the community, shows similar behaviour to *P. macrosturtensis* (B) and *P. mesosturtensis* (M) in the SIA (S3 Table). All the three species show similar dietary preferences in mixing models across the rainfall periods (S4 Table). While diets were dominated by amphipods AM1 and AM2 during the LR period (B: 39.9%, M: 49.3% and S: 47.9% (Fig 7)), predation/scavenging of sister beetle species accounted for the biggest dietary proportions during the wet season (B: 52.9%; M: 49.4%; S: 41.9% (Fig 7)).

Mixing models indicate that *P. macrosturtensis* larvae (Blv), which showed the biggest shift in trophic position, has a preference for amphipods *S. axfordi* (AM1) and *Y. sturtensis* (AM2) under LR conditions (accounting for 52% of the diet contributions), but also consumes a

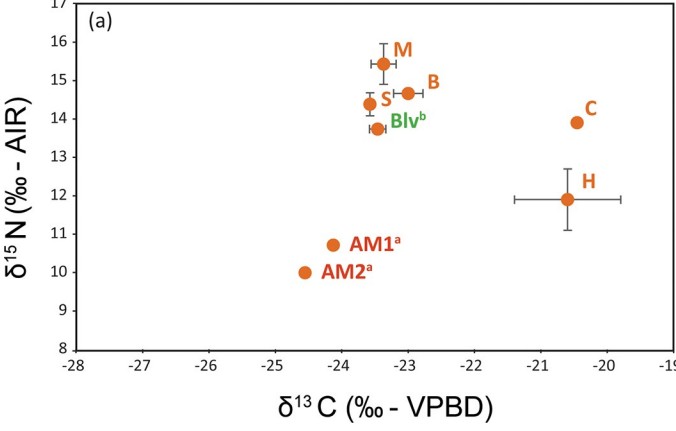
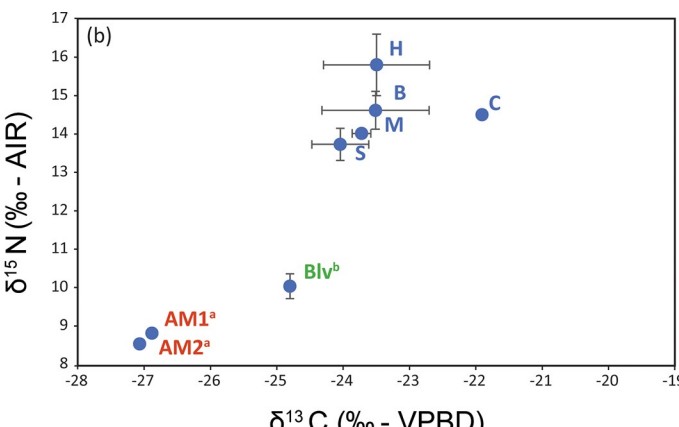

**Fig 5.** SIA biplots of adults *P. macrosturtensis* (B), *P. mesosturtensis* (M), *P. microsturtensis* (S), *P. macrosturtensis* larvae (Blv), *S. axfordi* (AM1), *Y. sturtensis* (AM2), Cyclopoida sp. (C) and Harpacticoida sp. (H) under low rainfall (a) and high rainfall (b). AM1[a] and AM2[a] (in red): taxa showing the biggest depletion in δ¹³C values for essential amino acids (phenylalanine and valine) across rainfall conditions; Blv[b] (in green): taxon showing the biggest drop in trophic position value (TP) between LR and HR. Refer to S3 Table for δ¹³C and δ¹⁵N values of the taxa.

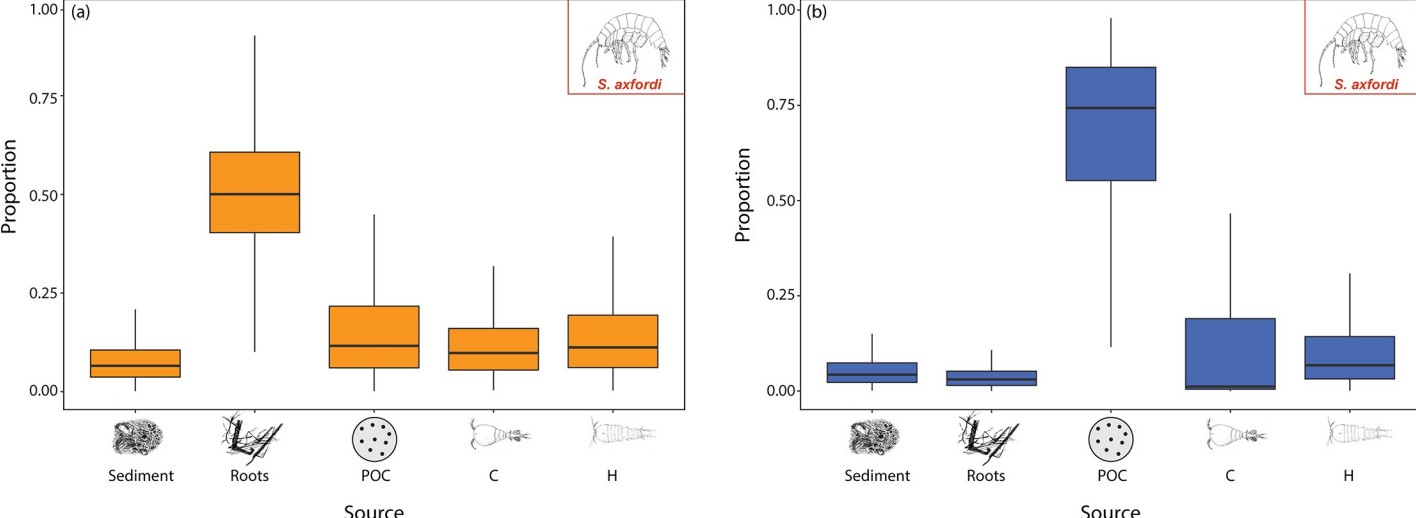

**Fig 6.** Modelled contributions to the diet of amphipod *S. axfordi* (AM1) under a) LR and b) HR conditions. POC: particulate organic carbon, C: Cyclopoida sp.; H: Harpacticoida sp. Medians and quartiles of each prey category are represented in the boxplot, see S3 Table for SIA $\delta^{13}$C and $\delta^{15}$N data. AM2 illustrated the same dietary preferences as AM1 under both rainfall conditions.

range of other organisms (Fig 8). During HR, Blv's diet is dominated by the two amphipod species, accounting for 79.6% of food sources (Fig 8). Overall, these results indicate changes in amphipods (AM1 and AM2) diet preferences linked with different OM inputs, coupled with enhanced species-specific predatory pressures from Blv under HR conditions.

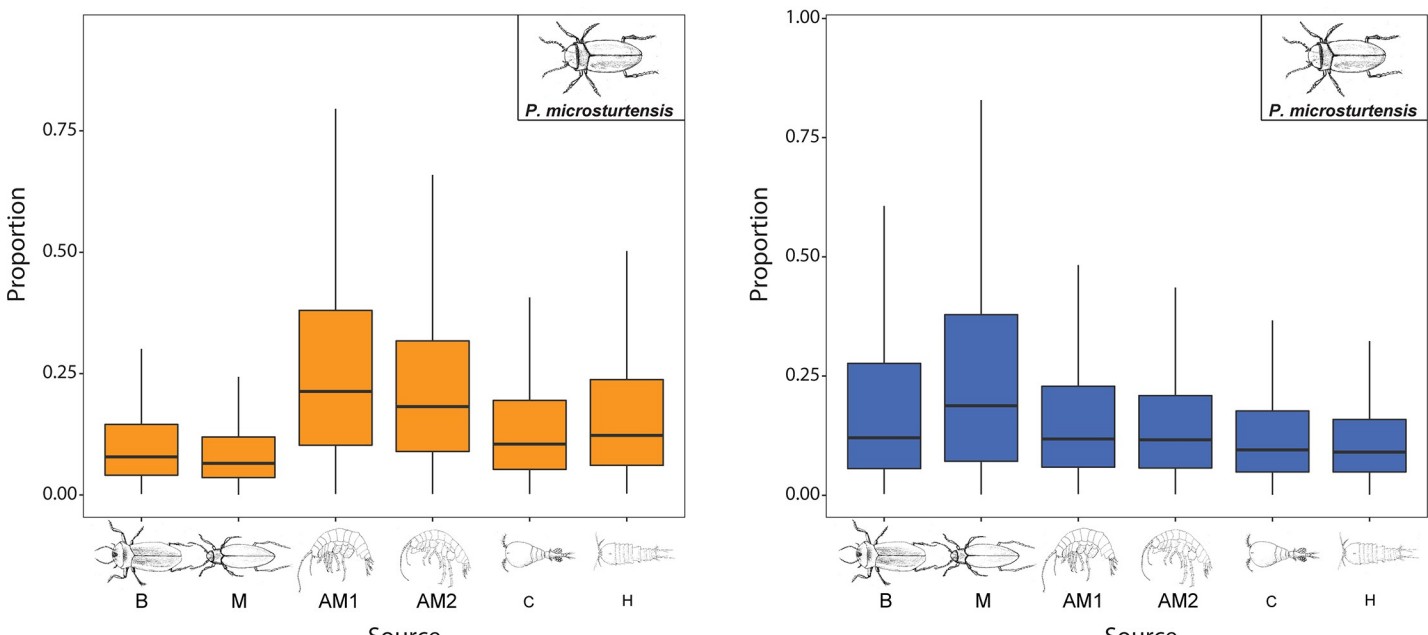

**Fig 7.** Contributions of *P. microsturtensis* adults' diet for a) LR and b) HR. Diet sources: *P. macrosturtensis* (B), *P. mesosturtensis* (M), *S. axfordi* (AM1), *Y. sturtensis* (AM2), Cyclopoida sp. (C) and Harpacticoida sp. (H). Medians and quartiles of each prey category are represented in the boxplot, see S3 Table for $\delta^{13}$C and $\delta^{15}$N bulk data. *P. macrosturtensis* (B) and *P. mesosturtensis* (M) illustrated same trends of dietary contributions across rainfall periods (S4 Table). In these analyses, sister species *P. mesosturtensis* (M) and *P. microsturtensis* (S) were considered as *Paroster* prey items for diet reconstruction of *P. macrosturtensis* (B), while contributions from *Paroster* diet sources *P. macrosturtensis* (B) and *P. microsturtensis* (S) were used for *P. mesosturtensis* (M).

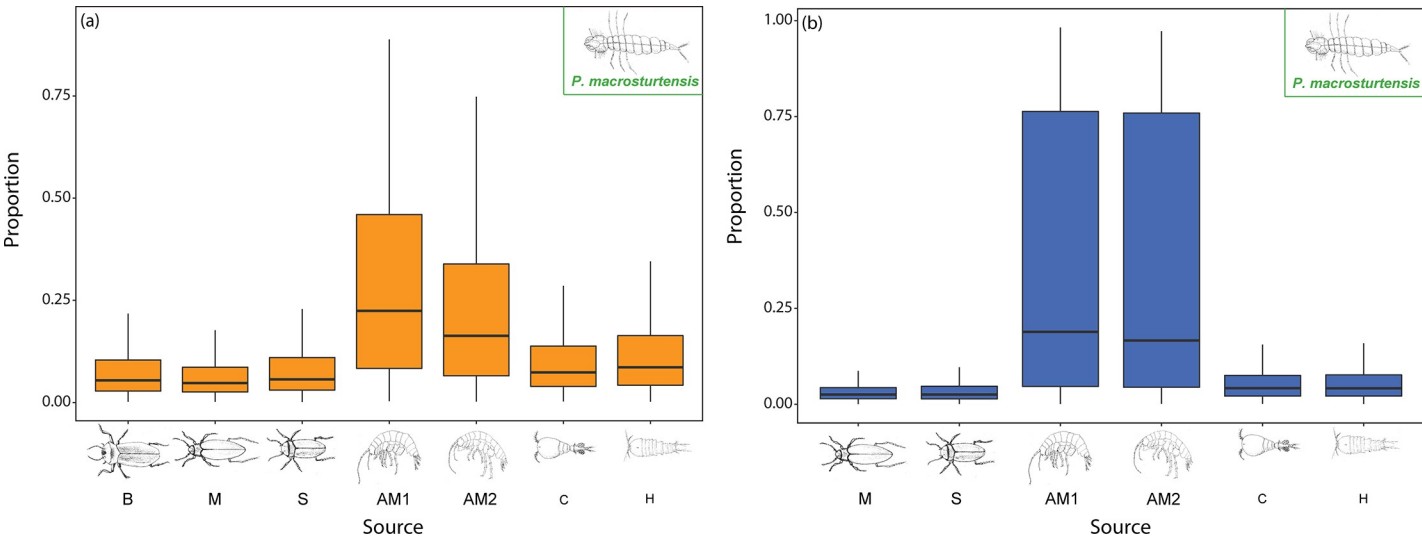

**Fig 8.** Stygofaunal contributions to the diet of *P. macrosturtensis* larvae for a) LR and b) HR. Diet sources: *P. macrosturtensis* (B), *P. mesosturtensis* (M), *P. microsturtensis* (S) *S.axfordi* (AM1), *Y. sturtensis* (AM2), Cyclopoida sp. (C) and Harpacticoida sp. (H). During HR, diet source *P. macrosturtensis* (B) was discarded as the Gelman-Rubin diagnostic reported values exceeding the corresponding upper confidence limits at the 95% confidence level. Medians and quartiles of each prey category are represented in the boxplot, see S3 Table for $\delta^{13}$C and $\delta^{15}$N bulk data.

## Discussion

### Shifts in basal OM assimilation

$\delta^{13}C_{EAA}$ data suggest that the stygofaunal community at Sturt Meadows experienced a seasonal shift in carbon flows during the wet season (HR). The overall tendency towards more $^{13}$C-depleted $\delta^{13}C_{Val}$ and $\delta^{13}C_{Phe}$ values indicate general stygofaunal discrimination against $^{13}$C sources in that season.

Three groups (OL, AM3 and S) showed a counter trend of $^{13}$C enrichment in the EAAs during HR. Of these, the easiest to account for are the oligochaetes (OL) which also showed increased abundances ($\chi^2$ = 6.7698, P < 0.05) of individuals ranging from 2 (LR1) and 1 (LR2) to 21(HR) (S1 Table), indicating ideal conditions for the taxon during the wet season. As detritovores, oligocheates may be expected to preferentially consume more degraded, and so $^{13}$C-enriched, OM. The enrichment in *S. bradfordae* (AM3) and *P. microsturtensis* (S) is harder to explain at this stage. The low abundance of *S. bradfordae* (AM3) means that, like the oligochaetes, it was not included in the SIA analysis and less data is available. This taxon, together with *P. macrosturtensis* larvae (Blv), *Y. sturtensis* (AM2) and water mites (OR), lacked statistically significant differences when $\delta^{13}$C values of EAA during LR are compared with those of NEAA involved in the glycolytic cycle. Newsome et al. [50] indicated that $\delta^{13}$C values of NEAA from diets of omnivorous animals reflect *de novo* synthesis but also dietary incorporations. Differential routing of macromolecules by consumers [55] are one possible contributor to our results. However, to date isotopic routing hypotheses have been tested only in vertebrates [30,56], with the study of metabolic pathways in aquatic invertebrates largely unexplored. Further CSIA investigations involving species-specific bio-assimilation processes within the stygofaunal community are needed to provide a more accurate understanding of the biochemical dynamics regulating this system.

In line with our general data trends, Hartland et al. [15], who reported consistent depletion in $\delta^{13}$C stygofaunal values within OM-enriched groundwaters *via* sewage contamination, concluded that stress-subsidy gradients in groundwaters trigger profound changes in stygofaunal

assemblages and have the potential to trigger shifts in feeding habits. Rainfall events trigger OM inflows which constitute high quality carbon sources for aquatic biota in groundwaters [16, 57].

Reiss et al. [26] demonstrated a strong link between nutrient inputs (mainly DOC) and groundwater microbe functional and metabolic richness after a major flooding event. Unfortunately, their methodology did not allow for corresponding macrofaunal trends to be identified. Nonetheless, microbially-derived OM incorporation by stygofauna has been reported in a number of groundwater ecology studies [16,21,58], and the biochemical importance of this linkage is widely accepted.

A key role in the observed trends at Sturt Meadows is played by amphipods which, together with copepods, are recognised as crucial actors in transferring OM to the upper stygofaunal trophic levels [16]. Specialized trophic habits in amphipods include epigean predation [10], detritivory [59], parasitism [60], biofilm grazing [61] and necrophagy [62]. Several studies have reported high degrees of trophic opportunism [54] and plasticity [63], allowing amphipods crucial shifts in feeding modes. Concurrently, niche partitioning has been addressed as a key mechanism to reduce intraspecific competition in ecosystems shaped by scarce nutrient availability [64]. However, our results do not show any conclusive evidence of epigean amphipod niche partitioning, with amphipods *S. axfordi* (AM1) and *Y. sturtensis* (AM2) showing the same dietary patterns. Overall, the isotopic data support the concept of opportunistic behaviours linked with changes in resource availability as a result of different rainfall regimes.

The HR event triggered substantial changes in the dietary proportions of *S. axfordi* (AM1) and *Y. sturtensis* (AM2), with notable decreases in root input and increases in POC. The extent of direct plant matter consumption by stygobionts reported in the literature–particularly by amphipods, which are facultative shredders–is both site and species-specific. Jasinska et al. [65] found that aquatic root mats were a key food source for a biodiverse cave fauna hosted by a shallow groundwater stream in Western Australia. Conversely, Navel et al. [66], reported the widely distributed amphipod species *Niphargus rhenorhodanensis* having preferential OM collector/gatherer feeding habits. In another study, Simon et al. [67] suggested that wood inputs played a role as indirect source of OM consumed by the ephilitic microbial mats which were ultimately targeted by common *Gammarus* amphipods.

At Sturt Meadows, a plausible explanation for the patterns observed is that during the dry season epigean amphipods rely on a more omnivorous diets where roots falling from the surface, and associated microbial and fungal biota, provide a substantial food source. Conversely, the wet season triggers inflows of replenished carbon ($^{13}$C-depleted POC) that might fuel biological turnovers in microbiological activity, and POC-attached microflora may be ultimately targeted by epigean amphipods. These assumptions are in line with the finding reported by Brankvotis et al. [16], and support the concept that grazers play a crucial role in sustaining the functional diversity in groundwaters. The importance of plant matter input during at least part of the year is supported by a previous bulk SIA investigation at Sturt Meadows [40], which also suggested that terrestrial sources of carbon, mainly DOC, reached the aquifer *via* percolation and play a crucial role in energy flows within the system. It is worth noting that our $\delta^{13}$C values of decarbonated sedimentary fractions (referred above as 'sediment') were less $^{13}$C-depleted than those in other groundwater investigations ([68,69,70], and had ranges close those for dissolved organic carbon (DIC) in the region ([71,72,73]. Portillo et al. [74] reported karst microbial growth induced by both carbonate precipitation and dissolution, suggesting the inclusion of inorganic carbon within the estimation of global carbon budgets in groundwaters. In line with this work, Chapelle [75] reported *in situ* DIC production as a result of microbial metabolism involved in the dissolution of carbonate material in the black Creek aquifer (California, USA). Our results suggest that carbonate assimilation and/or dissolution processes are likely

to occur in the sedimentary deposits of the aquifer, transferring an inorganic carbon isotopic fingerprint into the decarbonated and organic fractions. This can be tested in future by further functional studies on the microbial community [76] at the site.

Copepods (C and H) showed high $\delta^{15}N$ values compared to the rest of the stygofaunal community (S3 Table), suggesting alternative nitrogen sources linked to different microbial baselines. Copepods act as energy drivers in recycling nitrogen via ingestion of sediment and attached bacteria [12], with ammonia ($NH_3$), together with nitrates ($NO_3^-$), being an essential nutrient and energy source for subterranean microorganisms [77]. At Sturt Meadows aquifer, where ammonia levels are considerably higher than the natural concentrations [38], proliferation of selectively grazed ammonia-oxidising bacteria (AOB) might have played a key role in triggering the enriched $\delta^{15}N$ values in copepods.

The present study is constrained by its focus on stygofauna and therefore cannot provide direct evidence of microbially-derived ecological shifts. Future research needs to combine stygofaunal and microbial investigations to create a complete picture of the ecosystem. CSIA and functional genetic studies on microbes and copepods would also help define transitions from microflora to stygofauna, a process that has been understudied so far. Recent promising investigations in surface terrestrial [34] and aquatic [78] environments suggest a design—carbon fingerprinting—based on the incorporation of isotopic data into multifactorial mixing models that allow specific elucidation of bacterial sources in diets. Overall, despite the methodological challenges posed by groundwaters, isotopic data on stygofaunal carbon fluxes provides baseline knowledge that help untangle the intricate biochemical dynamics regulating subsurface environments.

### Trophic interactions

Our data on nitrogen CSIA pinpointed two main trophic levels marked by a small but clear separation between the top predators—adult beetles (B, M and S)—and primary consumer amphipods (AM1, AM2 and AM3) under both rainfall conditions. Compared to other ecosystems [29], the Sturt Meadows aquifer shows a very simple and truncated trophic web dominated by omnivorous habits. This is consistent with previous assumptions [67] due to the lack of primary producers [9] and scarce nutrient availability [79].

Within subterranean beetles, the smallest species *P. microsturtensis* (S), together with *P. macrosturtensis* (B), sit at the top of the trophic chain during HR (Table 3). Under those conditions, increased oxygen levels [38] may play a role in shaping changes in stygofaunal niche occupation. Subterranean beetles' body size has been found to drive differential physiological responses to increased exoskeleton respiration rates (inversely proportional to the body size) which ultimately affect the ability to allocate energy for breeding and foraging [80]. As the smallest species *P. microsturtensis* (S) can adapt their metabolism more quickly than direct competitor sister species *P. mesosturtensis* (M) under favourable conditions—such as HR regimes–they are more likely to show shifts in ecological niche occupation [38]. This trend, combined with the group feeding tendency of *P. microsturtensis* (S) beetles [40], indicates higher efficiency in activating more intensified predatory strategies when compared to *P. mesosturtensis* (M).

Dytiscidae beetle larval stages—commonly referred as 'water tigers'—are ferocious carnivores [81] with extremely opportunistic feeding behaviours involving scavenging and cannibalism [82]. At Sturt meadows, the third instar of blind *P. macrosturtensis* larvae (Blv) has a considerably bigger head capsule—paired with elongated mouthparts—than adult stages (Fig 9). These morphological features are likely to provide ethological advantages for non-visual predacious habits within light-less environments such as groundwaters [83]. This is consistent

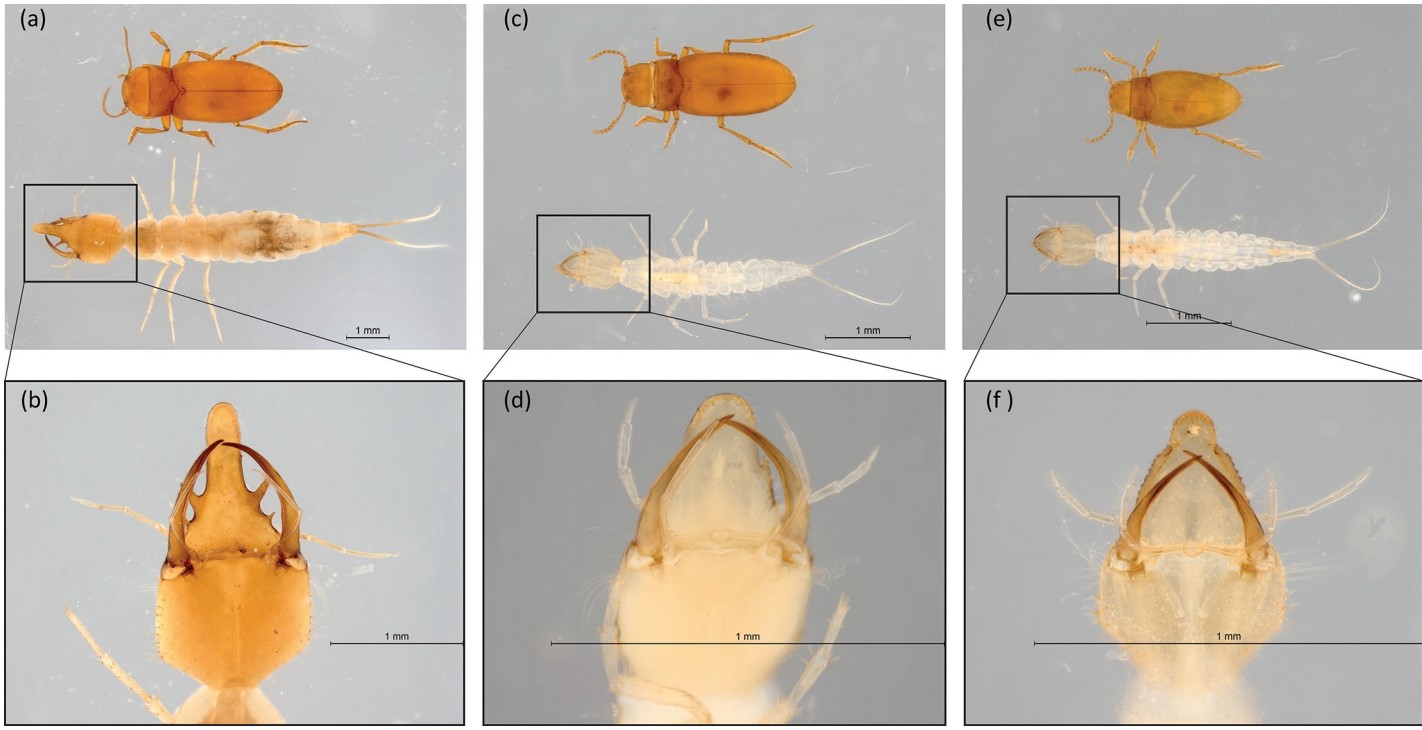

**Fig 9.** Comparisons between adult and larvae (whole body and head capsule) of *P. macrosturtensis* (a and b), *P. mesosturtensis* (c and d) and *P. microsturtensis* (e and f).

with stable isotope data from LR conditions positioning Blv at the top of the trophic web (Table 3).

Overall, modelled dietary contributions of *P. macrosturtensis* larvae (Blv) indicated a preference for amphipods (AM1 and AM2) coupled with residual cannibalism/scavenging (B, M and S) and predation of copepods (C and H) (Fig 8). Under HR, *P. macrosturtensis* larvae (Blv) showed the biggest drop in TP compared to LR ($TP_{LR-HR}$ = -0.3), which can be explained by an increased predatory focus on amphipods, and reduced feeding on secondary consumer sister species.

Previous work on surficial Dytiscidae larval stages published by Inoda et al. [84] stressed the importance of prey recognition through smell. According to their results, prey density was found not to be a key factor in shaping feeding behaviours, and self-other recognition played a role in group feeding. Overall, these findings indicated prevention of cannibalism through scent recognition. In groundwater, with total darkness and high influence of OM inputs on population dynamics [85,86], these patterns are likely to be strengthened. We suggest that the shifts in Blv predation seen in our results are dictated by a combination of chemical recognition and increased likelihood of encountering prey (amphipods) driven by enhanced resource availability (OM) during HR periods.

The role of bottom-up vs top-down forces in natural communities has been a cornerstone issue in the field of trophic ecology since the first empirical investigations [87]. Despite the controversy generated by the debate, there is now consensus that both forces act simultaneously on populations. This reinforces the need for whole system studies considering the interaction between heterogeneous (biotic and abiotic) forces and their effect on communities [88,89,90].

Our study, in line with a number of other investigations in the field [26,91] confirms that rainfall events *via* water advection are key drivers in defining energy flows and ecological

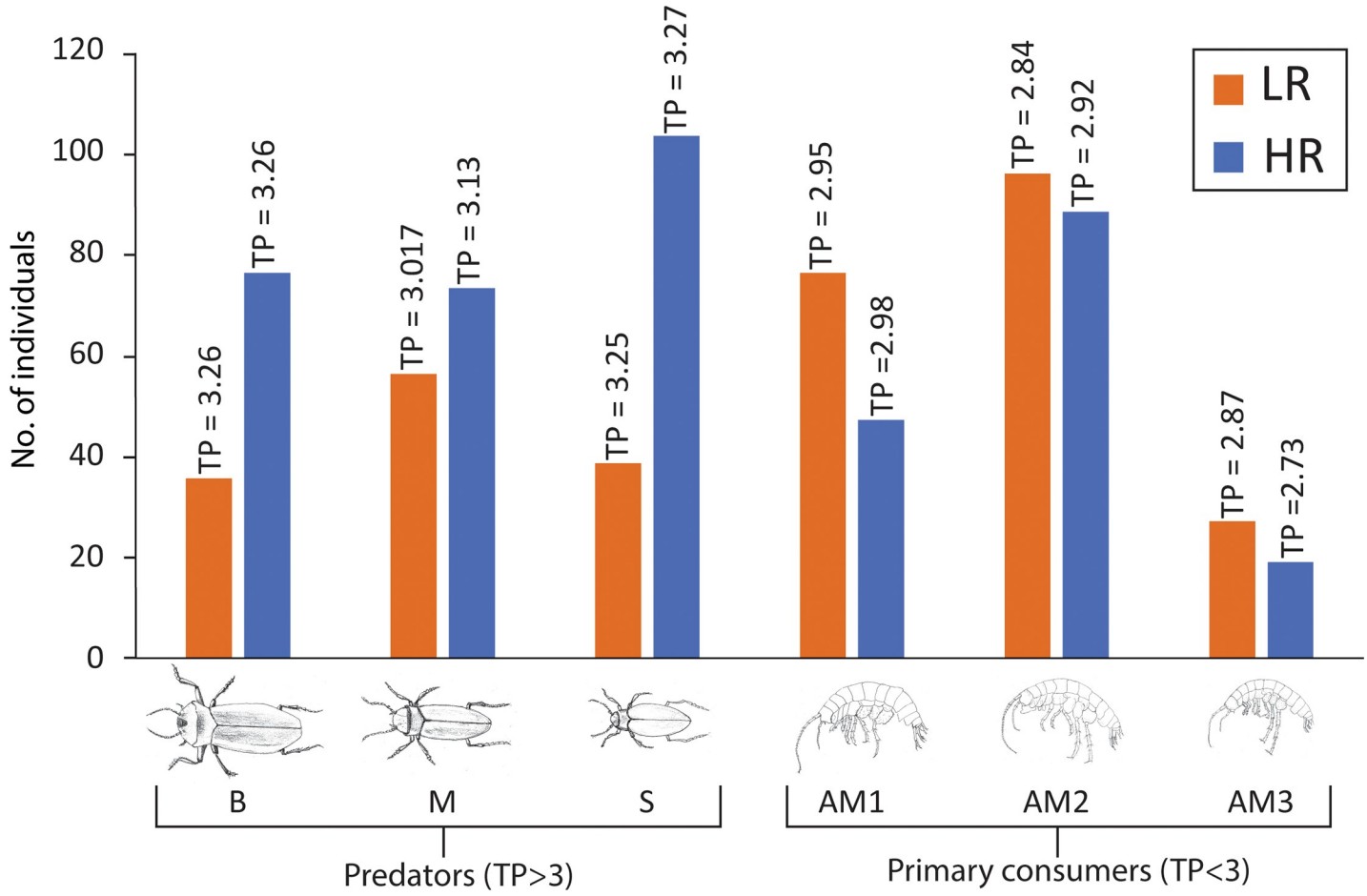

**Fig 10. Bar chart graphs comparing dry season abundances (as the average value of LR1 and LR2) with HR conditions for top predators (beetles B, M and S) and key prey items (amphipods primary consumers AM1, AM2 and AM3).** See S1 Table for detailed abundance data. None of the abundances of these taxa changed significantly between LR and HR. See Saccò et al. [38] for detailed statistical analyses across LR1, LR2 and HR.

patterns within resource-limited environments. We suggest that OM-driven bottom-up regulations, increasingly accepted as driving factors shaping population dynamics in aquifers [92], shaped the shifts in feeding behaviours among amphipod taxa in the calcrete. However, despite the beneficial conditions for primary consumers triggered by increased nutrient availability (i.e. microbial biofilms) and better environmental settings (i.e. increased oxygen, [38]), a decrease in amphipod populations under HR indicates the existence of additional ecological factors.

Top-down forces (i.e. natural predators), widely studied in surface aquatic ecosystems [93,94], have hardly been addressed in groundwater. Previous genetic investigations at Sturt Meadows pinpointed predatory pressures from beetles on amphipods and copepods [95] and reported a lack of trophic niche partitioning among the *Paroster* species. In another study, Hyde [96] reported evidence from metagenomics data suggesting that subterranean blind beetles at Sturt Meadows feed on both prey invertebrates and their sister species. Our isotope results support these hypotheses, indicating opportunistic predaceous habits in the calcrete, mixed with scavenging/cannibalism. However, substantial uncertainty remains about the magnitude of interspecific predatory pressures among *Paroster* sister species, and further species-specific lab experiments are needed to investigate these ethological aspects.

Biochemical functional role interpretation coupled with abundance data suggests that bottom-up population dynamics are counterbalanced in the system by top-down forces. Increased numbers of top predators (adult beetles B, M and especially S) were paired with a decrease of key prey items (amphipods AM1, AM2 and AM3) when HR is compared with the dry season (LR1 and LR2) (Fig 10).

In light of the dynamics shown by our isotope data, we suggest that the reported shift in amphipods (AM1 and AM2) carbon incorporation during HR might have triggered changes in their ecological behaviour, exposing them to increased predatory pressures from the top predator *Paroster* beetles (B, M and S). However, given the high degree of opportunistic behaviour reported by stygofauna [14], further investigations on species-specific ethological dynamics would be helpful to infer community dynamics.

The number of third instar dytiscidae larvae 'Blv' did not vary across sampling campaigns, suggesting differential ecological niche occupations. Previous investigations on *Paroster* larvae detected three instars before pupating, with the first two occupying a reduced proportion of their lifetime [83]. Future investigation of early stages of larval developments are needed to establish if potential population blooms (i.e. mass reproduction) are linked with contrasting recharge periods.

## Conclusions

The application of CSIA and SIA allows elucidation of the trophic dynamics shaping stygofaunal communities in an arid zone calcrete aquifer. Rainfall acts as a driver in regulating both top down and bottom up changes in dietary habit. Subterranean invertebrate population dynamics are notoriously hard to investigate due to sampling obstacles and a current lack of knowledge around stygofaunal biological cycles [7,36]. However, our isotopic results allow a greater insight into the food web dynamics and the biogeochemical forces that shape them than has previously been possible. Further investigations involving higher numbers of samples from more biodiverse systems or complex trophic assemblages (i.e. alluvial aquifers) will help refine the approach. The incorporation of qualitative analyses such as DNA metabarcoding would also complement quantitative isotopic methods to provide crucial insights into processes (i.e. cannibalism) and key driving forces (i.e. niche partitioning) that are hard to detect *via* one method alone. Lastly, investigation of nitrogen sources and their isotopic changes would open up the nitrogen data collected to interpretation beyond trophic position.

Groundwater environments are fundamentally important to ecosystems, communities and industry, and a robust understanding of their ecosystem dynamics is essential to accurately assess environmental impacts, whether anthropogenic, or climatic. Isotopic data, especially if combined in multidisciplinary studies with other parameters [22] has a key role to play in elucidating previously hard to investigate function within these cryptic systems.

## Supporting information

**S1 Table. Abundance data of the different stygofaunal taxa at Sturt Meadows detected during the sampling campaigns LR1, LR2 and HR.**
(XLSX)

**S2 Table. Sample weights (mg) for SIA and CSIA analyses.** n/a: not available.
(XLSX)

**S3 Table. δ13C and δ15N values of stygofauna, roots, sediment and POC during LR and HR.**
(XLSX)

**S4 Table. Dietary proportions of *P. macrosturtensis* (B), *P. mesosturtensis* (M) and *P. microsturtensis* (S) under LR and HR conditions.**
(XLSX)

## Acknowledgments

We wish to acknowledge the traditional custodians of the land, the Wongai people, and their elders, past, present and emerging. We acknowledge and respect their continuing culture and the contribution they make to the life of Yilgarn region. The authors acknowledge NERC group at Bristol University for the Life Sciences Mass Spectrometry Facility (LSMSF) and Dr Alice Mora for the support with the facilities at La Trobe Institute for Molecular Sciences (LIMS). The authors thank Flora, Peter and Paul Axford of the Meadows Station are thanked for their kindness and generosity in providing both accommodation and access to their property.

## Author Contributions

**Conceptualization:** Alison J. Blyth, William F. Humphreys.

**Data curation:** Mattia Saccò, Alison J. Blyth, Alison Kuhl, Debashish Mazumder, Colin Smith.

**Formal analysis:** Mattia Saccò, Alison Kuhl.

**Funding acquisition:** Alison J. Blyth, William F. Humphreys.

**Investigation:** Mattia Saccò, William F. Humphreys.

**Methodology:** Mattia Saccò, Colin Smith.

**Project administration:** Alison J. Blyth.

**Resources:** Alison J. Blyth, Debashish Mazumder, Colin Smith, Kliti Grice.

**Supervision:** Alison J. Blyth, William F. Humphreys, Kliti Grice.

**Validation:** William F. Humphreys, Alison Kuhl, Debashish Mazumder, Colin Smith, Kliti Grice.

**Writing – original draft:** Mattia Saccò.

**Writing – review & editing:** Alison J. Blyth, William F. Humphreys, Kliti Grice.

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
