## [Decision Letter · Decision Letter 0]

4 Sep 2019

[EXSCINDED]

PONE-D-19-21542

Elucidating stygofaunal trophic web interactions via isotopic ecology

PLOS ONE

Dear Mr. Sacco,

Thank you for submitting your manuscript to PLOS ONE. After careful consideration, we feel that it has merit but does not fully meet PLOS ONE’s publication criteria as it currently stands. Therefore, we invite you to submit a revised version of the manuscript that addresses the points raised during the review process.

Please pay attention specially to explain or provide a reason for the high sediment d13C values in your study.

We would appreciate receiving your revised manuscript by Oct 19 2019 11:59PM. To enhance the reproducibility of your results, we recommend that if applicable you deposit your laboratory protocols in protocols.io, where a protocol can be assigned its own identifier (DOI) such that it can be cited independently in the future. For instructions see: http://journals.plos.org/plosone/s/submission-guidelines#loc-laboratory-protocols

We look forward to receiving your revised manuscript.

Kind regards,

Jose M. Riascos, Ph.D.

Academic Editor

PLOS ONE

Journal Requirements:

1. In your Methods section, please provide additional information regarding the permits you obtained for the work. Please ensure you have included the full name of the authority that approved the field site access and, if no permits were required, a brief statement explaining why.

Reviewers' comments:

Reviewer's Responses to Questions

**Comments to the Author**

1. Is the manuscript technically sound, and do the data support the conclusions?

Reviewer #1: Yes

Reviewer #2: Yes

2. Has the statistical analysis been performed appropriately and rigorously? 

Reviewer #1: I Don't Know

Reviewer #2: Yes

3. Have the authors made all data underlying the findings in their manuscript fully available?

Reviewer #1: Yes

Reviewer #2: Yes

4. Is the manuscript presented in an intelligible fashion and written in standard English?

Reviewer #1: No

Reviewer #2: Yes

5. Review Comments to the Author

Reviewer #1: This article shows a great deal of field and laboratory work, where a multi-disciplinary approach is used to elucidate a complex series of questions regarding trophic interactions and their shifts within stygofauna species comparing two distinct environmental seasons. Bulk and compound specific stable isotope analysis coupled with environmental factors are linked together to explain a series of changes in stygofauna composition, feeding behavior and trophic shifts. The authors do a good job justifying their research and stating the importance of groundwater ecosystems and the stygofauna. They present a clear and concise background on the role that stygofauna play in the carbon cycle and how it incorporates into the trophic web, setting the stage for the methods the authors selected to answer the research questions. The objectives of this work are stated clearly.

It would make this paper a lot easier to read if the authors would include a broader description of precisely how d13C CSIA on EAA and NEAA allows the identification of source carbon signatures (lines 81-86), giving a hint to what the reader should expect to find and thus, would make the reader fully understand the differences pointed out in the results. As a suggestion, including what the authors expected to find on CSIA EAA and NEAA would also be nice.

Their data supports a change in feeding habits of several stygofaunal components from different levels of trophic web, thus demonstrating the importance of carbon input in these subterranean ecosystems.

Overall, the text is well structured, the results are very interesting and I believe this article would serve as a good baseline for future work in Australian calcretes. However, to my understanding, some of the assertions in the discussion are not fully supported by the data and some of the results are not fully discussed. Thus, with some adjustments, I would recommend this article to be published.

Annotations and suggestions:

Line number: Comment.

Methods: Some specifications would help the reader to have a better understanding of exactly how the sampling took place and would be vital for replication: How deep below the surface is the water table in this region? Does it vary throughout the year? Are 11m enough to get deep into the groundwater? How far is the distance from the water table to the bottom? how much does the net go underwater?

78: It is not clear if the authors will use SIA to refer to stable isotope analysis or to bulk material Stable isotope analysis.

127: The authors should state how the boreholes were selected (randomly or selected and the selecting criteria). Could there be any bias in the boreholes selected?

131: In an effort to describe a replicable method, please comment on the methods used to obtain such samples: were the sediments collected from the bottom of the borehole, how were they obtained and what instruments were used?

133: The authors should mention why they consider that these 2 boreholes are representative of the rest and how they were chosen.

145: Figure caption requires modification of EEA to EAA as is referred to in the figure. Initials in figure need to be explained: LC-iRMS (liquid chromatography- Isotope Ratio Mass Spectrometry), GC-iRMS (Gas), EA-iRMS (Elemental analysis)

161: Possibly the authors would use the above defined abbreviation?

164-166: This procedure is missing or unclear in Fig. 2

228: It would help the reader know the number of specimens (n) the authors used in the different sets of analysis.

234: Tukey

272-277: I would suggest the authors to be consistent with the abbreviations, probably using NEAA and EAA would eliminate the need for a new abbreviation in this table.

Check for misspellings in Table 1.

Table 1. Annotation 2: using only Ser, Gly and Ala is not explicitly explained or justified in the text by the authors and is suggested.

279: The results shown here should be further explained and clarified to suggest patterns of predators or prey so the readers understand exactly what the authors are pointing out.

286-293: This section of results was very nicely explained.

303: Table 2: The readers would appreciate the taxon names along with the ID´s

320: the authors should identify which is LR and HR within a and b in the caption of fig4.

325: I would suggest the authors be consistent with the description style and include the taxon name before the abbreviation ID, as they did in 232-235.

326: A sample size number (n) would help the readers have an idea of what is a low abundance in these sites, it could help the authors to remember it may be very different in other subterranean environments.

336: table 3 caption: It would help the reader to know the n size in each of the samples, the authors should also specify what the range after each value is referring to (SD, error, variance, etc). The code description under the TP pairwise comparison column is missing in the caption.

350: the authors should use formal English.

356: VPBD and AIR are not explained in caption of figure. I also suggest the authors to include 13C and 15N values of sources in their figures, it can give a better understanding to the reader of how these sources relate to each of the stygofaunal elements and how these may also shift during LR and HR.

367: I would suggest the authors include the number of specimens obtained.

368: LR has been defined as dry season previously.

368-377: It appears as this paragraph has been separated or is unclear and the authors need to revise it.

370: AM1 is mentioned in the text and figure caption but AM2 is shown in the figure. Are the differences shown significantly different?

377: It would help the reader if this was mentioned along with the reference to fig 6.

379-381: it would help the reader if the authors would include the taxon name again.

381: The authors are suggested to include the n values in table S1.

382: table S2 has not been mentioned and the authors are invited to consider including it into their results, considering it will be discussed further ahead and the abundance data (and figure) provide interesting information on the community assemblage.

387: misspelling

388: C and H are not used in this figure.

396: There is no sustaining evidence from the authors to claim a change in amphipod behavior in HR; the authors should explain what the amphipod ethology is in LR and HR (and if it differs among species), and further explain how these changes -coupled with increased OM inputs- trigger the species specific predatory response from the P. mesosturtensis larvae (Blv).

398: There appear to be several things to note in figures 7 & 8: S is not present in figure 7, which could mean P. microsturtensis has no cannibalistic behavior (although mentioned earlier) and that other beetles are not feeding on S either (if "B and M illustrated same trends" as noted in fig 7.

B is not present in fig8.b, possibly meaning that Blvs do not feed on adults of their same species during HR. I would suggest the authors to mention these differences in the results section to avoid misinterpretation from the readers.

420-421: POC and DOC would hint the readers to think of carbon instead of matter (and they were also defined earlier in the text).

416-427: Apparently reference [50] says the epilithic biofilms are consumed preferentially from the OM that enters from the surface, and refers to the “high quality” carbon from outside as an assumption, but I could not find their quality measurements in this reference. I would suggest the authors to consider looking at J. Pohlman´s work (1997, 2011) and Brankovits et al 2017 [16] for additional information on "quality" carbon and other production sources in flooded caves. An alternative, would be to consider in situ production, such as it has been observed in other subterranean ecosystems and would probably correspond to a “groundwater microbe functional and metabolic richness” increment after OM recharge event, as demonstrated by [27].

435-437: Although it is not this papers scope, is it possible there was an abundance difference within these species among your monitored sites? If so, could these differences be suggesting intraspecific competition?

449-451: Including the sources 13C and 15N in the figures shown above, could help the reader follow this description.

452: This statement could confuse the readers. Would the authors please expand their explanation on how 13C depleted POC (sources with a greater abundance in 12C) is fresh carbon? Did the authors mean 14C, the replenishment of C sources or something else?

481: Authors should clarify whether these results are from another study or include their environmental data in the supporting information.

483-484: I would suggest the authors to mention that these amazing beetles actually breathe through their exoskeleton. Some readers are unfamiliar with the Australian stygofauna.

479-489: I suggest the authors are consistent throughout the document when referring to the taxon or ID´s of the species.

500: which species are the authors referring to?

501: This statement is confusing to the reader, it appears to suggest that feeding on another predator is less nutritious than feeding on a primary consumer, is there supporting evidence?

502: please include the ID code in the taxon box within figure 8, since the text refers to the code instead of the name.

502-504: this sentence is a little confusing, it would help the reader if the authors would be more specific in their description of when the increased predatory focus occurred (Hr or LR); Consistency of abbreviations would be clearer: "under HR, Blv...compared to the LR"

505-510: In this paragraph it is not clear when the authors are referring to data from [66] and when they are addressing their own findings.

510-513: The authors have not made a statement of any estimation of prey density in HR or LR and there is no reference to this data. This is unsupported. -Figure 10 and table S2 need to be mentioned earlier; probably in methods or results. Statistical analysis should be provided for this claim.

519-527: As a reader who potentially does not know the particular conditions of this site, is it possible that the authors could consider an autochtonous (in situ) primary production that could fuel the trophic web throughout the LR as observed in other studies (16)? –probably a good place in the manuscript to discuss?-

528: Do the authors refer to natural predators?

529: References 77 and 78, though doctoral dissertations, have not been through a peer reviewed process and thus should be used carefully. -78 reference year is 2018, not 2010.

547: Being this the only plausible explanation exposed, I would suggest the authors expand their explanation and present or point to supporting information that leads to their hypothesis.

Questions that strike me are tied to their biological reproductive cycle (for both prey and predators), such as discussed below.

569: authors should revise the sentence.

Reviewer #2: Sacco et al. use stable isotopes of carbon and nitrogen (both bulk and amino acids) to unravel trophic structure of a stygofaunal community. They find clustering of organisms around trophic level 3 and report a change in source use between a low and high rainfall period. The paper is generally well written and the topic is interesting. I have a few comments I hope will improve the manuscript.

1. I think the intro or methods could use a bit more description of the conditions in the boreholes. How far does light penetrate? Should we expect some primary production in the surface waters that could sink to the bottom, or is it entirely runoff from the surface? Are the copepods that were sampled from near the surface, or are they also in permanent darkness? What about POC samples (surface or dark)? How were the roots and sediment samples obtained, and what does the latter represent? Sediment is a mixture rather than a source. Adding detail on the source pathways will help contextualize the isotope data that come later.

2. The number of organisms captured should be expressed as per sampling unit (e.g. # of organisms per tow), since five tows were used at each site at each time. Comparing abundances at different times relies on standardized effort across sites and times. Also, the abundance data should probably be reported in the results section so it doesn’t come as a surprise later (i.e. before line 526 and Figure 10).

3. The sediment has unusually high d13C values, suggestive of carbonate contamination. It is noted that POC samples were acid washed. What about the sediments? Were they also acid washed? Without acid washing, I don’t trust the reported value and it could skew the mixing model very strongly (see Phillips et al. 2014 Can. J. Zool. 92: 823-835).

4. I find it interesting that none of the organisms analysed for CSIA came back as primary consumers. All were near TL = 3. So who are the primary consumers? This points to microbial conditioning of detrital sources as the likely pathway leading to metazoans. This should be highlighted in the discussion.

5. The mixing model did not resolve the diets of adult beetles particularly well (lots of overlap in 95% CIs), so this shouldn’t be stated as a significant difference between seasons.

Specific comments

Line 25 and elsewhere – when using the term “depleted”, always refer to the heavier isotope i.e. “13C-depleted”.

Line 78 – delete “SIA”

Line 84 – change “signatures” to “isotopes” since signature refers to large well-buffered reservoirs

Line 140 – delete “the”

Line 157 – I was surprised to hear the biofilm referred to as “shredding”. Is that an error? Do microbes shred?

Line 162 – are the copepods not considered to be stygofauna?

Line 320 – add (a) and (b) to the figure caption

Line 348 – it is worth noting here that copepods had very high d15N relative to the rest of the food web (not just in HR), despite being assumed to be primary consumers. This implies a distinct N source for these organisms (i.e. different baseline), and it is unfortunate that they weren’t analysed for CSIA. If they truly are primary consumers, then the CSIA would have revealed it.

Line 370 – add “mean and 95% credible interval” to the figure caption

Line 378 – are the depletions here referring to 13C or 15N or both?

Line 397 – mean proportions were the same in both seasons so this statement is rather speculative

Line 407 – suggest adding “in that season” after “sources”

Line 479 – this paragraph is very speculative and could be deleted without affecting the paper’s interpretations.

Line 496 – it might be worth noting that Blv were not at the top of the trophic web when bulk data were considered, perhaps owing to the different baseline d15N value of the copepods

Line 515 – field “of” trophic ecology

Line 569 – quantitative isotopic “methods”

6. PLOS authors have the option to publish the peer review history of their article (what does this mean?). If published, this will include your full peer review and any attached files.

Reviewer #1: Yes: Efrain M. Chavez Solis

Reviewer #2: No

---

## [Author Response · Author response to Decision Letter 0]

26 Sep 2019

Elucidating stygofaunal trophic web interactions via isotopic ecology

Ms. Ref. No.: PONE-D-19-21542

Title: Elucidating stygofaunal trophic web interactions via isotopic ecology

Journal: PLOS ONE

Please accept our sincere gratitude for providing challenging but highly constructive comments and suggestions, which, we believe, have greatly improved the paper.

Below, we provide a detailed, point-by-point account of how we responded to each comment

In yellow in the MS: changes according to suggestions from reviewer 1

In light blue in the MS: changes according to suggestions from reviewer 2

In green in the MS: authors’ contributions to an improved version of the manuscript 

Please pay attention specially to explain or provide a reason for the high sediment d13C values in your study.

Thank you for your comment. Sediment samples (decarbonated sedimentary fractions) were hydrolysed to remove the inorganic component. δ13C values of sediment are close to the ones of DIC in water, suggesting a linkage to carbonate metabolisms as indicated by Portillo et al., 2009. Naturwissenschaften, 96(9), 1035-1042. Further details on the explanation of the observed patterns are reported in the response to reviewer 2 (major comment 3.). We also improved grammar and composition along the manuscript.

1. In your Methods section, please provide additional information regarding the permits you obtained for the work. Please ensure you have included the full name of the authority that approved the field site access and, if no permits were required, a brief statement explaining why.

Thank you for your note. We included the permit number and the full name of the authority for the fieldwork activities (Lines 159-160).

Reviewer #1: This article shows a great deal of field and laboratory work, where a multi-disciplinary approach is used to elucidate a complex series of questions regarding trophic interactions and their shifts within stygofauna species comparing two distinct environmental seasons. Bulk and compound specific stable isotope analysis coupled with environmental factors are linked together to explain a series of changes in stygofauna composition, feeding behavior and trophic shifts. The authors do a good job justifying their research and stating the importance of groundwater ecosystems and the stygofauna. They present a clear and concise background on the role that stygofauna play in the carbon cycle and how it incorporates into the trophic web, setting the stage for the methods the authors selected to answer the research questions. The objectives of this work are stated clearly.

It would make this paper a lot easier to read if the authors would include a broader description of precisely how d13C CSIA on EAA and NEAA allows the identification of source carbon signatures (lines 81-86), giving a hint to what the reader should expect to find and thus, would make the reader fully understand the differences pointed out in the results. As a suggestion, including what the authors expected to find on CSIA EAA and NEAA would also be nice.

Thank you for your suggestion. We included a section in the Introduction (Lines 83-87) explaining the difference between EEA and NEAA and its implication for the carbon fingerprinting. In the methodology section, we also specified why we focus in EAA and referred to their potential of elucidating shifts in stygofaunal OM incorporations under different rainfall conditions (Lines 267-277). In addition, and for clarity purposes, we differentiated between NEAA involved into Krebs cycle and glycolysis. We incorporated new results in Table 1 and referred to them in the results (Line 321-326) and discussion (Lines 471-481), stressing the need to widen the knowledge about isotopic ecology patterns in invertebrates. 

Their data supports a change in feeding habits of several stygofaunal components from different levels of trophic web, thus demonstrating the importance of carbon input in these subterranean ecosystems.

Overall, the text is well structured, the results are very interesting and I believe this article would serve as a good baseline for future work in Australian calcretes. However, to my understanding, some of the assertions in the discussion are not fully supported by the data and some of the results are not fully discussed.

Thank you for your comment. We considerably improved the discussion and support the suggestions provided with bibliographic references (Lines 457-647).

Annotations and suggestions:

Line number: Comment.

Methods: Some specifications would help the reader to have a better understanding of exactly how the sampling took place and would be vital for replication: How deep below the surface is the water table in this region? Does it vary throughout the year? Are 11m enough to get deep into the groundwater? How far is the distance from the water table to the bottom? how much does the net go underwater?

Thank you for your suggestions. We incorporated information about the calcrete (Line 113-115). We also referred to the in press manuscript (Ecohydrology Journal) “Stygofaunal community trends along varied rainfall conditions: deciphering ecological niche dynamics of a shallow calcrete in Western Australia” (reference [38] in the manuscript) for further details about the sampling design, monitoring of water depth and hydrogeological background at Sturt Meadows (Lines 121-122).

78: It is not clear if the authors will use SIA to refer to stable isotope analysis or to bulk material Stable isotope analysis.

Thank you for your comment. We removed “SIA” in the Line 78 as suggested also by the reviewer 2 for clarity purposes.

127: The authors should state how the boreholes were selected (randomly or selected and the selecting criteria). Could there be any bias in the boreholes selected?

Thank you for your notes. We included information about the simple random criteria for the sampling (Line 137) and reference to the previously mentioned in press manuscript (reference [38] for further details). Overall, we included a sampling procedure that involves a third of the accessible and non-dry bores in situ. This extensive sampling effort provides a robust and reliable methodological approach that gives us confidence that the stygofaunal diversity and abundance ranges of the area are representatives and non-biased. 

131: In an effort to describe a replicable method, please comment on the methods used to obtain such samples: were the sediments collected from the bottom of the borehole, how were they obtained and what instruments were used?

Thank you for your notes. We included further details about sediment samples collection, separation and pre-treatment (Lines 142-146).

133: The authors should mention why they consider that these 2 boreholes are representative of the rest and how they were chosen.

Thank you for your comment. We included further detailed explanation on how these two bores where selected and why we consider them representative of the hydroecological groundwater dynamics (Lines 150-153).

145: Figure caption requires modification of EEA to EAA as is referred to in the figure. Initials in figure need to be explained: LC-iRMS (liquid chromatography- Isotope Ratio Mass Spectrometry), GC-iRMS (Gas), EA-iRMS (Elemental analysis).

Thank you for your notes. We replaced EEA with EAA in the Figure 2 and explained the analytical acronyms in the figure caption (Lines 172-175).

161: Possibly the authors would use the above defined abbreviation?

Thank you for your suggestion. We replaced the sentence with the above defined abbreviation (Line 190).

164-166: This procedure is missing or unclear in Fig. 2

Thank you for your comment. We included CF (continuous flow) in the acronym – and caption – of the Figure 2 (Line 174).

228: It would help the reader know the number of specimens (n) the authors used in the different sets of analysis.

Thank you for your notes. As reflected in the section “Samples preparation and study design”, all the individuals collected were pooled together and separated into LR and HR. For this reason, instead of considering the number of specimens, for clarity purposes we specified the range of mg per samples considered in each analysis in the text (Line 191,199, 208 and 228) and referred to a newly created supplementary table (Table S2) for further details. This approach is conventionally reported in many isotopic studies (Takano et al., 2010 Rapid Communications in Mass Spectrometry 24.16 (2010): 2317-2323; Chikaraishi et al., 2009 Limnology and Oceanography: methods, 7(11), 740-750) and allows comparison of analytical effort between different laboratories procedures and machine setups. 

234: Tukey

Thank you for your comment. We amended the spelling mistake (Line 274).

272-277: I would suggest the authors to be consistent with the abbreviations, probably using NEAA and EAA would eliminate the need for a new abbreviation in this table.

Thank you for your suggestion. We used NEEA and EAA for consistency and clarity purposes along the manuscript.

Check for misspellings in Table 1.

Thank you for your note. We amended the misspellings in the Table 1.

Table 1. Annotation 2: using only Ser, Gly and Ala is not explicitly explained or justified in the text by the authors and is suggested.

Thank you for your comment. We agree with the review that further information is required, and we considerably improved this section according to the suggestion. In the section titled ‘Data treatment and statistical analyses’ we provided details about the two final biochemical pathways (Krebs cycle: intermediates serine (Ser), glycine (Gly) and alanine (Ala); Glycolysis: intermediates) considered for the NEAA analyses and comparison to EAA (Lines 267-272). Results from the new comparison (EAA vs NEAA(Glycolisis)) were integrated in the Table 1, detailed in the results (Line 321-326) and discussed in the section ‘Shifts in OM basal assimilation’ (Lines 471-481).

279: The results shown here should be further explained and clarified to suggest patterns of predators or prey so the readers understand exactly what the authors are pointing out.

Thank you for your suggestion. These results refer to the average values of EAA and NEEA, and their comparison across the stygofaunal community studied. As detailed in the text, δ13C EAA average values were significantly different (and more depleted) compared to NEAA. Aside the mentioned exceptions, same patterns were observed for both preys and predators, so we did not include further explanation/categorisation as suggested. It is worth underlining that EAA are the main focus of the present study for the analysis of potential shifts in OM incorporations. For this reason, the suggested clarification and analysis were provided for EAA as reported in the section spanning from the line 327 to 334. We consider that this section, including the Figure 3 and the improved Table 2, provides clear displaying of the results linked to the EAA patterns and consequent OM shifts (further discussed in the Discussion section).

286-293: This section of results was very nicely explained.

Thank you for your note.

303: Table 2: The readers would appreciate the taxon names along with the ID´s

Thank you for your suggestion. We included the names and ID’s as per Table 1.

320: the authors should identify which is LR and HR within a and b in the caption of Figure 4.

Thank you for your note. We included (a) and (b) in the caption of the Figure 4 (Lines 360-361).

325: I would suggest the authors be consistent with the description style and include the taxon name before the abbreviation ID, as they did in 232-235.

Thank you for your suggestion. For consistency purposes, we ‘formatted’ the taxa descriptions along the MS as ‘species name’ followed by ‘(Sample ID)’. 

326: A sample size number (n) would help the readers have an idea of what is a low abundance in these sites, it could help the authors to remember it may be very different in other subterranean environments.

Thank you for your comment. We included the number of total individuals per each one of the two taxa (Lines 365-368), and referred to Table S1 (former Table S2) for the abundance data.

336: table 3 caption: It would help the reader to know the n size in each of the samples, the authors should also specify what the range after each value is referring to (SD, error, variance, etc). The code description under the TP pairwise comparison column is missing in the caption.

Thank you for your comment. We included details about the group lettering and the SD in the Figure caption for clarity purposes (Lines 376-378). As specified in the response to the comment about the Line 228, we refer to the mg of sample employed in each analysis instead of the total number of specimens for clarity purposes. The inclusion of the total number of individuals sampled in this table would expose the reader to the misleading assumption on the totality of the sampled individuals employed for nitrogen CSIA. For this reason, we are inclined not to incorporate the suggested data. However, for clarity purposes we increased the number of citations of the abundance Table (Table S1) along the text in order to avoid misunderstanding between abundance data and isotope samples.

350: the authors should use formal English.

Thank you for your comment. We replaced ‘couldn’t’ with ‘could not’ (Line 391).

356: VPBD and AIR are not explained in caption of figure. I also suggest the authors to include 13C and 15N values of sources in their figures, it can give a better understanding to the reader of how these sources relate to each of the stygofaunal elements and how these may also shift during LR and HR.

Thank for your notes. We detailed VPBD and AIR references in the methodology section (Lines 196-199). We refer to Table S3 in the figure caption for the specific values 13C and 15N values since we think that displaying the data on the graph would duplicate results.

367: I would suggest the authors include the number of specimens obtained.

Thank you for your suggestion. We included in brackets the average value for LR (between LR1 and LR2) and HR (Line 409).

368: LR has been defined as dry season previously.

Thank you for your note. We eliminated ‘dry season’ and phrased ‘during low rainfall conditions (LR)’ (Lines 410-411)

368-377: It appears as this paragraph has been separated or is unclear and the authors need to revise it.

Thank you for your notes. We included the expression ‘Regarding dietary preferences of the amphipod S. axfordi (AM1)’ (Line 410) to link this paragraph to the previous section.

370: AM1 is mentioned in the text and figure caption but AM2 is shown in the figure. Are the differences shown significantly different?

Thank you for your comments. There was an error in the figure. We corrected the name of the species (AM1, S. axfordi) in the Figure 6 and improved the description of the figure caption by reporting that AM2 illustrated the same trends as AM1 (Lines 419-420). Bayesian models do not employ P values but work with posterior probability distributions. As detailed in the methodology section (Lines 303-305), MCMC convergence was tested via Gelman-Rubin diagnostic (with the conventional threshold of 1.1) to investigate the fit of the posterior predictive model. Please refer to Parnell et al., 2013 Environmetrics, 24(6), 387-399 for further details.

377: It would help the reader if this was mentioned along with the reference to fig 6.

Thank you for the suggestion. We included the sentence ‘AM2 illustrated the same dietary preferences as AM1 under both rainfall conditions’ in the caption of Figure 6 (Lines 419-420).

379-381: it would help the reader if the authors would include the taxon name again.

Thank you for your suggestion. We included the taxon names for consistency purposes See response to the comment on the Line 325).

381: The authors are suggested to include the n values in table S1.

Thank you for your suggestion. Please see the previous responses to the comment about line 228 and 336.

382: table S2 has not been mentioned and the authors are invited to consider including it into their results, considering it will be discussed further ahead and the abundance data (and figure) provide interesting information on the community assemblage.

Thank you for your comment. We referred to the abundance data in the Table S1 (former Table S2) in the methodological section (Line 138) so it does not come as a surprise later on the manuscript (as suggested also by the Reviewer 2). However, we are inclined to leave the Figure 10 for the discussion since it mixes both abundance and newly generated isotopic data (TP). Given our study has the main focus to illustrate ecological patterns via isotopic approaches, we consider that Figure 10 provide useful insights that have to be discussed in the discussion section of the manuscript for consistency and clarity purposes. 

387: misspelling

Thank you for your note. We amended the misspelling.

388: C and H are not used in this figure.

Thank you for your comment. We homogeneised the coding for copepods and harpacticoids along the manuscript in the figure and figures (Lines 398-399,417 and 434-435). For clarity purposes, we employed the codes (C and H) in the figures and refereed to their taxa Cyclopoida sp. (C) and Harpacticoida sp. (H) in the figure caption. Supplementary Tables included either both the ID code and taxonomic name (Table S1) or just the code ID (Table S2 and Table S3) for copepods (cyclopoids and harpacticoids).

396: There is no sustaining evidence from the authors to claim a change in amphipod behavior in HR; the authors should explain what the amphipod ethology is in LR and HR (and if it differs among species), and further explain how these changes -coupled with increased OM inputs- trigger the species specific predatory response from the P. mesosturtensis larvae (Blv).

Thank you for your comment. We restructured the sentence and eliminated the ethological suggestion (Lines 447-449). Beyond the evident shift from roots to POC, no further information about changes in amphipods’ ethology are available to date. As a result, in this section we limited our dissertation to the outcomes of the Bayesian modelling, and left further suggestion/potential explanations for the final discussion. 

398: There appear to be several things to note in figures 7 & 8: S is not present in figure 7, which could mean P. microsturtensis has no cannibalistic behavior (although mentioned earlier) and that other beetles are not feeding on S either (if "B and M illustrated same trends" as noted in fig 7.

B is not present in fig8.b, possibly meaning that Blvs do not feed on adults of their same species during HR. I would suggest the authors to mention these differences in the results section to avoid misinterpretation from the readers.

Thank you for your comment. Figure 7 refers to the dietary proportions of P. microsturtensis (S) and, among the other Paroster species, included only sister species B and M for two main reasons. First of all, no evidence of adult cannibalism (i.e. S adults actively predating/scavenging on other S adults) has been reported in previous studies [40]. With this in mind, we included ‘(and potential active predatory pressures) on sister species’ in the Line 183 to clarify that molecular evidence indicates interspecific - and not intraspecific - interactions. Second, the investigation of cannibalistic behaviours on adult beetles is out of the scope of our manuscript for methodological/analytical reasons. While several studies focuses on isotopic techniques to investigate cannibalistic behaviours, the vast majority of them employ feeding/ethological experiments in the lab (e.g. Venturelli et al., 2006. Transactions of the American Fisheries Society, 135(6), 1512-1522). In these mesocosm experiments, individuals from the same species are usually put in strict contact for a certain amount of time and anomalous nitrogen enrichments are used as an indicator of cannibalistic behaviour. Our study focuses on the functioning of trophic interactions at community level and did not incorporate mesocosm experiments because of both the bias that artificial environments can provide and the lack of genetic data supporting the cannibalistic hypothesis at Sturt Meadows. Indeed, we agree that additional information is key to understand the whole extent of the stygofaunal trophic interactions at Sturt Meadows, and we advocate for further analysis able to bring insights into these complex dynamics (i.e. Lines 636-641 in the discussion).

We also restructured the sentence in the caption of the Figure 7 (Lines 433-441) by specifying that B and M illustrated the same trends of dietary proportion with the exclusion of cannibalism (we considered Paroster prey items M and S for B’s diets, and Paroster prey items B and S for M’s diets).

Stygofaunal contributions to the diet of P. macrosturtensis larvae (Blv) did not consider cannibalism (Blv feeding on B) during HR because the Gelman-Rubin diagnostic reported a higher number than the conventional 1.1 threshold, indicating a not reliable posterior predictive model for P. macrosturtensis adults (B). For clarity purposes, we included this explanation in the caption of the Figure 8 (Lines 452-454). 

420-421: POC and DOC would hint the readers to think of carbon instead of matter (and they were also defined earlier in the text).

Thank you for the We removed the expression in brackets for clarity purposes (Line 484).

416-427: Apparently reference [50] says the epilithic biofilms are consumed preferentially from the OM that enters from the surface, and refers to the “high quality” carbon from outside as an assumption, but I could not find their quality measurements in this reference. I would suggest the authors to consider looking at J. Pohlman´s work (1997, 2011) and Brankovits et al 2017 [16] for additional information on "quality" carbon and other production sources in flooded caves. An alternative, would be to consider in situ production, such as it has been observed in other subterranean ecosystems and would probably correspond to a “groundwater microbe functional and metabolic richness” increment after OM recharge event, as demonstrated by [27].

Thank you for your comment. We included the Pohlman, 2011 ([57]) reference and removed the fomer reference [50] from the sentence for consistency purposes. We referred to the in situ production later on along the discussion (Lines 526-537 and 538-545) as we agree that might play a key role in these systems.

435-437: Although it is not this papers scope, is it possible there was an abundance difference within these species among your monitored sites? If so, could these differences be suggesting intraspecific competition?

Thank you for your notes. Interesting point. However, as reported in the MS in press on Ecohydrology (Saccò et al., 2019b) [38]: ‘The community was not distributed differently across the five geological areas and the number of individuals, taxa, Shannon and Evenness indexes did not change significantly according to the geological zones across the different rainfall periods’. Other abundance-based preliminary analyses (unpublished data) also discarded conclusive information about inter and intraspecific competition patterns. Given the opportunism that these species manifest, we argue that other techniques such as eDNA or CSIA could provide useful insights into ecological patterns otherwise very difficult to detect in groundwater systems (Saccò et al., 2019a)[22]. 

449-451: Including the sources 13C and 15N in the figures shown above, could help the reader follow this description.

Thank you for your suggestion. We consider that reporting the isotopic values of the sources would break the flow of the discussion. d13C and d15N values are reported in the Table S3 and its reference is mentioned in the caption of the figures mentioned by the reviewer (Figure 5,6,7,8). 

452: This statement could confuse the readers. Would the authors please expand their explanation on how 13C depleted POC (sources with a greater abundance in 12C) is fresh carbon? Did the authors mean 14C, the replenishment of C sources or something else?

Thanks for your notes. Since we refer to the replenishment of C sources, we used the expression ‘replenished carbon’ in the text (Line 519).

481: Authors should clarify whether these results are from another study or include their environmental data in the supporting information.

Thank you for the comment. We referred to the in press MS (Saccò et al., 2019b)[38] reporting the information referred in the text. 

483-484: I would suggest the authors to mention that these amazing beetles actually breathe through their exoskeleton. Some readers are unfamiliar with the Australian stygofauna.

Thank you for your suggestion. We included the suggested information in the Line 569.

479-489: I suggest the authors are consistent throughout the document when referring to the taxon or ID´s of the species.

Thank you for your suggestion. For consistency purposes, we ‘formatted’ the taxa descriptions along the MS as ‘species name’ followed by ‘(Sample ID)’.

500: which species are the authors referring to?

Thank you for your comment. We refer to P. macrosturtensis larvae (Blv) and we improved the sentence for clarity purposes (Line 587).

501: This statement is confusing to the reader, it appears to suggest that feeding on another predator is less nutritious than feeding on a primary consumer, is there supporting evidence?

Thank for your comment. We agree with the reviewer. We removed the part ‘(and less nutritious)’ (Line 588) 

502: please include the ID code in the taxon box within figure 8, since the text refers to the code instead of the name.

Thank you for your comment. We improved the Figure 8 by referring to Cyclopoida sp. and Harpacticoida sp as ‘C’ and ‘H’, same as per Figure 7.

502-504: this sentence is a little confusing, it would help the reader if the authors would be more specific in their description of when the increased predatory focus occurred (Hr or LR); Consistency of abbreviations would be clearer: "under HR, Blv...compared to the LR"

Thank you for your comment. We improved the sentence according to the reviewer’s suggestions (Lines 587-592).

505-510: In this paragraph it is not clear when the authors are referring to data from [66] and when they are addressing their own findings.

Thank you for your comment. We added ‘According to their results’ for clarity purposes (Line 594).

510-513: The authors have not made a statement of any estimation of prey density in HR or LR and there is no reference to this data. This is unsupported. -Figure 10 and table S2 need to be mentioned earlier; probably in methods or results. Statistical analysis should be provided for this claim.

Thank you for your notes. Table S1 (former Table S2) is mentioned in the methods section for clarity purposes. However, we are inclined to leave the Figure 10 for the discussion section as explained in the response to the comment on the Line 382. Our suggestion in the (former) lines 510-513 does not involve any consideration of the prey/predator abundances across the rainfall periods, but it rather focuses on biochemically-based observations. In fact, this section of the manuscript (former 510-513) focuses on the interpretation of trophic shifts by considering the isotopical data, presented in the result section, composed by Blv’s dietary contributions, TP estimations and amphipods’ (AM1 and AM2) shifts in OM incorporations. For this reason, we consider that this suggestion is supported and pertinent to the discussion of the results. Taxa abundances were not significantly different between LR (mean between LR1 and LR2) and HR and we included this information in the figure caption. We also referred to Saccò et al., (2019b) [38] for further details about the analysis of abundance patterns along LR1, LR2 and HR (Lines 633-635).

519-527: As a reader who potentially does not know the particular conditions of this site, is it possible that the authors could consider an autochtonous (in situ) primary production that could fuel the trophic web throughout the LR as observed in other studies (16)? –probably a good place in the manuscript to discuss?-

Thank you for your suggestion. For clarity purposes, we included information about potential in situ production in two different sections within the paragraph ‘Shifts in basal OM assimilation’. First, microbial carbonates incorporations/production are inferred from the d13C fingerprints of the sediment samples (Lines 526-537). Second, we suggest alternative nitrogen sources linked to different microbial baselines potentially involving ammonia oxidation metabolisms (Line 538-545). 

528: Do the authors refer to natural predators?

Thank you for your note. We replaced ‘enemies’ with ‘predators’ (Line 616).

529: References 77 and 78, though doctoral dissertations, have not been through a peer reviewed process and thus should be used carefully. -78 reference year is 2018, not 2010.

Thank you for your comment. We agree with the reviewer that conclusions from not peer reviewed works should be treated carefully. However, we consider that the two cited works provide interesting insights that improve the understanding of the ecological processes at the studied aquifer. Moreover, being groundwater processes notoriously site-specific, the cited doctoral dissertation provides vital genetically-based information that contribute to comprehension of the trophic dynamics at Sturt Meadows.

547: Being this the only plausible explanation exposed, I would suggest the authors expand their explanation and present or point to supporting information that leads to their hypothesis.

Questions that strike me are tied to their biological reproductive cycle (for both prey and predators), such as discussed below.

Thank you for your notes. We restructured the sentence by stressing that this suggestion is in light of the dynamics described along the manuscript (Line 636). In addition, we have used caution in inferring ethological changes in amphipods, and referred to the ecological changes rather than purely behavioural/biological patterns. We also removed any reference to changes in biological fitnesse for clarity and consistency purposes (Lines 636-641). 

569: authors should revise the sentence.

Thank you for your comment. We included ‘methods’ (Line 658) as suggested by the reviewer 2.

Reviewer #2: Sacco et al. use stable isotopes of carbon and nitrogen (both bulk and amino acids) to unravel trophic structure of a stygofaunal community. They find clustering of organisms around trophic level 3 and report a change in source use between a low and high rainfall period. The paper is generally well written and the topic is interesting. I have a few comments I hope will improve the manuscript.

1. I think the intro or methods could use a bit more description of the conditions in the boreholes. How far does light penetrate? Should we expect some primary production in the surface waters that could sink to the bottom, or is it entirely runoff from the surface? Are the copepods that were sampled from near the surface, or are they also in permanent darkness? What about POC samples (surface or dark)? How were the roots and sediment samples obtained, and what does the latter represent? Sediment is a mixture rather than a source. Adding detail on the source pathways will help contextualize the isotope data that come later.

Thank you for your comments. We included further description of the boreholes (Lines 117-118). Light does not penetrate since the bores are capped. In addition, the natural habitat in the calcretes is in permanent darkness and there is nowhere access to light. Spatial information about primary production on site is not available to date and it is limited to results from functional genomics involving proteobacteria (Hyde et al., 2018). Copepods were sampled via haul nets along the water columns, same as per the rest of stygofauna, and therefore no further explanation is provided. Further explanation of the POC sampling methodology is provided (Lines 153-156) for clarity purposes. Roots and sediment were collected through the haul netting procedure (Lines 142-144), the latter representing the particulate sediment from the bottom of the aquifer as stated in the Line 142. We also referred to the in press manuscript titled “Stygofaunal community trends along varied rainfall conditions: deciphering ecological niche dynamics of a shallow calcrete in Western Australia” (Saccò et al., (2019b)) [38] for further details about sampling design, monitoring of water depth and hydrogeological background at Sturt Meadows (Lines 121-122).

2. The number of organisms captured should be expressed as per sampling unit (e.g. # of organisms per tow), since five tows were used at each site at each time. Comparing abundances at different times relies on standardized effort across sites and times. Also, the abundance data should probably be reported in the results section so it doesn’t come as a surprise later (i.e. before line 526 and Figure 10).

Thank you for your comments. Abundance data (Table S1, former Table S2) is referred in the methods section, earlier in the manuscript for clarity purposes (Line 138). Standardised procedures, by one operator, were carried out to maximise the representativeness of the sampling methodology. Five hauls, the most reliable sampling approach at Sturt meadows as reported by Allford et al. (2005), were repeated for every sampled bore to retain specimens along the water column (and not just in the upper layer in close proximity to the borehole case). As a result, the sampling unit considered is composed by 5 hauls and it is not possible to separate the single tows as suggested. While we concord on the necessity to dig into representativeness of sampling procedures in groundwater environments, we also consider that the methodology applied - with the technology and knowledge available to date - provides the rigour and robustness required. After sampling the aquifers over more than 15 years now we have found a general consistency with which bore holes are associated with different taxa and how abundant these taxa are (see Hyde et al. 2018; when taking account of seasonal effects), which provides us with some confidence that our sampling methods are reliable and allow a comparison of relative abundance patterns over time.

3. The sediment has unusually high d13C values, suggestive of carbonate contamination. It is noted that POC samples were acid washed. What about the sediments? Were they also acid washed? Without acid washing, I don’t trust the reported value and it could skew the mixing model very strongly (see Phillips et al. 2014 Can. J. Zool. 92: 823-835).

Thank you for your comment. As per POC samples, sediment samples were hydrolysed to remove the inorganic component. These samples indicated d13C values in the range of dissolved inorganic carbon values in water as reported by several studies in groundwater environments (Lipar et al., 2017. Palaeogeography, palaeoclimatology, palaeoecology, 470, 11-29; Cartwright et al., 2007. Australian Journal of Earth Sciences, 54(8), 1103-1122; Rowe et al., 2000. Palaeogeography, Palaeoclimatology, Palaeoecology, 157(1-2), 109-125.), suggesting that the organic component of the sediment (lipids, sugars, microbes and biofilm) is strictly related to carbonate metabolisms. This is not unusual, as widely detailed by Portillo et al., 2009. Naturwissenschaften, 96(9), 1035-1042. Indeed, we agree with the Reviewer 2 that further CSIA carbon fingerprinting analyses are necessary to unravel the carbon assimilatory pathways in the system. However, provided the whole-system focus of our work, we consider that the improved methodology (Line 117-160), results (i.e. Table 1, Lines 321-326) and discussion (Line 471-481; Lines 526-545) sections on OM incorporations provided in the MS allow untangling of the main biochemical processes shaping Sturt Meadows calcrete aquifer.

4. I find it interesting that none of the organisms analysed for CSIA came back as primary consumers. All were near TL = 3. So who are the primary consumers? This points to microbial conditioning of detrital sources as the likely pathway leading to metazoans. This should be highlighted in the discussion.

Thank you for your comment. Very interesting point. Our isotopically-based data suggest that copepods act as pure primary consumers at Sturt Meadows. We agree with the reviewer that this outcome indicates linkages between primary consumers and microbes. For this reason, in the paragraph ‘Shifts in basal OM assimilation’ we included two sections about potential DIC microbial metabolisms (as suggested by the low d13C values of the decarbonated sediments, Lines 526-537) and the linkage copepods-microbes as suggested by the high d15N values of copepods and harpacticoids under both rainfall regimes (Lines 538-545). Unfortunately, methodological constraints prevented the analysis of CSIA on copepods. Once combined with functional genetic studies, these biochemically-based results will bring further light to the crucial linkage copepods-microbes (Line 546-556).

5. The mixing model did not resolve the diets of adult beetles particularly well (lots of overlap in 95% CIs), so this shouldn’t be stated as a significant difference between seasons.

Thank you for your note. We corrected the wrong reference to the 95% CIs by replacing it with the correct description “Medians and quartiles of each prey category are represented in the boxplot” in the captions of the figures 6 (Line 418), (Lines 435-436) and 8 (Lines 454-455). As a result, the box plots presented in this study do not refer to potential overlapping in dietary proportion. While we agree with the reviewer that the overall diet of P. microsturtensis (S) is quite similar under the two rainfall periods (LR and HR), mixing models revealed considerably different contributions from amphipods (AM1 and AM2) and sister species of beetles (B and M) under LR and HR. We are therefore inclined to report the observed differences due to the fact that they provide interesting insights into trophic shifts (amphipods-based diet vs sister species predation/scavenging) under contrasting rainfall periods.

Specific comments

Line 25 and elsewhere – when using the term “depleted”, always refer to the heavier isotope i.e. “13C-depleted”.

Thank you for your comment. We added “13C-“ to “depleted” along the manuscript.

Line 78 – delete “SIA”

Thank you for your note. We deleted “SIA” (Line 74).

Line 84 – change “signatures” to “isotopes” since signature refers to large well-buffered reservoirs

Thank you for your suggestion. We changed “signatures” to “isotopes”.

Line 140 – delete “the”

Thank you for your comment. We deleted “the” (Line 164).

Line 157 – I was surprised to hear the biofilm referred to as “shredding”. Is that an error? Do microbes shred?

Thank you for your comment. Indeed, it is an error. We restructured the sentence for clarity purposes. (Lines 186-187).

Line 162 – are the copepods not considered to be stygofauna?

Thank you for your comment. We deleted “copepods” for clarity purposes (Line 190). Meiofauna was considered as stygofauna as suggested by the reviewer.

Line 320 – add (a) and (b) to the figure caption

Thank you for the comment. We added (a) and (b) to the figure caption (Lines 360-361).

Line 348 – it is worth noting here that copepods had very high d15N relative to the rest of the food web (not just in HR), despite being assumed to be primary consumers. This implies a distinct N source for these organisms (i.e. different baseline), and it is unfortunate that they weren’t analysed for CSIA. If they truly are primary consumers, then the CSIA would have revealed it.

Thank you for your comments. We agree with the reviewer that copepods revealed anomalously high d15N values compared to rest of stygofauna. As detailed in the response to the main point 4, we discussed about the different baseline and potential metabolisms implied in the discussion section (Lines 538-545). Indeed, CSIA on groundwater copepods would provide key information due to their crucial role in linking microbes and upper stygofaunal taxa. Hopefully, future technological advances will allow analysis of even smaller amount of sample, allowing better understanding of the energy flows in groundwaters.

Line 370 – add “mean and 95% credible interval” to the figure caption

Thank you for your suggestion. We included “Medians and quartiles of each prey category are represented in the boxplot” for the captions of Fig.6, 7 and 8, since the intervals represented refer to the quartiles instead of the 95% credible intervals (and medians instead of means). See response to the main point 5 for further details.

Line 378 – are the depletions here referring to 13C or 15N or both?

Thank you for your note. We added “δ13C and δ15N” for clarity purposes (Line 425).

Line 397 – mean proportions were the same in both seasons so this statement is rather speculative

Thank you for your comment. We included the amphipods’ proportions (AM1 and AM2) under LR (52% in the Blv diet) to the underline their increase under HR (80%) (Line 445). We also restructured the following sentence without considering ethological changes in amphipods AM1 and AM2 for clarity purposes (Lines 447-449).

Line 407 – suggest adding “in that season” after “sources”

Thank you for the suggestion. We added “in that season” for clarity purposes (Line 463).

Line 479 – this paragraph is very speculative and could be deleted without affecting the paper’s interpretations.

Thank you for your comment. Further reference to the in press manuscript “Stygofaunal community trends along varied rainfall conditions: deciphering ecological niche dynamics of a shallow calcrete in Western Australia” (Saccò et al., (2019b))[38] has been added to support the increase in oxygen levels under HR (Line 567) and also sustain the S’s shifts in ecological niche occupations (Line 573). Moreover, we included a new reference to the study carried out by Bradford et al., 2013 [40] when S’s group feeding is mentioned (Line 574). Given all these updates, we consider that this paragraph is much less speculative and can provide interesting insights into S’s ecological patterns.

Line 496 – it might be worth noting that Blv were not at the top of the trophic web when bulk data were considered, perhaps owing to the different baseline d15N value of the copepods

Thank you for your comment. We are inclined to think that biochemical routing could have played a role in shaping this SIA result. Moreover, given CSIA provides a much more accurate pinpointing of the trophic positions along the food chain, we used it as reference and avoided mentioning any SIA-based trophic level characterisation for consistency purposes. Copepods played a marginal role in Blv’s diets (Figure 7) and provided the complexity of the biochemical flows along the stygofaunal community. We consider that further analyses are required to attribute the suggested linkage to different baseline assimilations.

Line 515 – field “of” trophic ecology

Thank you for your note. We included “of” (Line 603).

Line 569 – quantitative isotopic “methods”

Thank you for your note. We included “methods” (Line 658).

---

## [Editor Report · Decision Letter 1]

3 Oct 2019

Elucidating stygofaunal trophic web interactions via isotopic ecology

PONE-D-19-21542R1

Dear Dr. Sacco,

We are pleased to inform you that your manuscript has been judged scientifically suitable for publication and will be formally accepted for publication once it complies with all outstanding technical requirements.

With kind regards,

Jose M. Riascos, Ph.D.

Academic Editor

PLOS ONE
---

## [Editor Report · Acceptance letter]

7 Oct 2019

PONE-D-19-21542R1 

Elucidating stygofaunal trophic web interactions via isotopic ecology 

Dear Dr. Sacco:

I am pleased to inform you that your manuscript has been deemed suitable for publication in PLOS ONE. Congratulations! Your manuscript is now with our production department. 

With kind regards,

on behalf of

Professor Jose M. Riascos 

Academic Editor

PLOS ONE